# Analytical framework for evaluating NMPC-based robot navigation in fluid environments

Maram Ali[1,2], Saptarshi Das[1,3]*, Stuart Townley[1,4]

1 Centre for Environmental Mathematics, Faculty of Environment, Science and Economy, University of Exeter, Penryn, Cornwall, United Kingdom, 2 College of Science, King Khalid University, Abha, Saudi Arabia, 3 Institute for Data Science and Artificial Intelligence, University of Exeter, Exeter, Devon, United Kingdom, 4 Environment and Sustainability Institute, University of Exeter, Penryn, Cornwall, United Kingdom

* saptarshi.das@ieee.org

## Abstract

This paper presents a novel hybrid simulation framework by combining nonlinear model predictive control (NMPC) and the lattice Boltzmann method (LBM) for autonomous robot navigation in dynamic environments. We evaluated the control algorithm's resilience by looking at fluid-structure interactions in both laminar ($Re = 100$) and turbulent ($Re = 2000$) flows. To ensure numerical accuracy and physical fidelity, a systematic grid independence study was conducted across various resolutions ($10 \times 10$ to $200 \times 200$). The $200 \times 200$ grid was selected as the benchmark standard, providing 3–5 lattice units within the viscous boundary layer to minimize numerical diffusion and accurately resolve high-frequency vortex shedding patterns. This rigorous validation allowed us to test the NMPC trajectory planning across fundamentally different flow behaviors with high confidence in the underlying hydrodynamics. The aim is to enhance mobile robot navigation by integrating a resilient control algorithm with a comprehensive fluid dynamics study, focusing on enhancing trajectory planning, obstacle avoidance, and overall performance in dynamic fluid environments. Computational fluid dynamics (CFD) analysis is combined with a robust control algorithm. The robot's interaction with the surrounding fluid is evaluated through different parameters such as Reynolds number, drag forces, the robot's energy dissipation, and vorticity. Key performance metrics, including a path efficiency of 0.887 and low computational requirements with the LBM-NMPC framework maintaining a linear memory footprint of 2.88 MB at peak resolution, demonstrate NMPC algorithm's viability as a fast and efficient trajectory planner. The robot maintained safe distances from obstacles, highlighting the effectiveness of the obstacle avoidance strategy and the robustness of the validated simulation environment.

**Data availability statement:** All relevant data are within the paper and its Supporting information files.

**Funding:** The work of Maram Ali was supported by King Khalid University and the Saudi Arabia Cultural Bureau in the UK. The funders had no role in study design, data collection and analysis, decision to publish, or preparation of the manuscript.

**Competing interests:** The authors have declared that no competing interests exist.

# 1 Introduction

Dynamic obstacle avoidance is a critical problem for autonomous robots that has gained significant attention due to its applications in areas such as industrial automation, autonomous vehicles, and intelligent navigation systems [1,2]. Developing robust methods to ensure reliable robot navigation in environments with static and dynamic obstacles while considering external influences, such as fluid interactions, remains a challenging task. This paper presents a hybrid approach that combines the nonlinear model predictive control (NMPC) method and the lattice Boltzmann method (LBM) to address this problem effectively. NMPC framework has been widely used for trajectory tracking and obstacle avoidance in robotics due to its ability to handle nonlinear dynamics and constraints explicitly. NMPC optimizes control inputs by predicting future states over a finite time horizon, making it an appropriate option for real-time applications in dynamic environments [3,4]. However, standard NMPC controllers for undersea vehicles frequently depend on simpler kinematic or dynamic models, which are insufficient for accurately representing intricate fluid phenomena such as vortex shedding and dynamic drag. This creates a significant gap: whereas a controller may excel in a simplified simulation, its actual efficacy in a complicated fluid environment is unverified. This research addresses this gap by introducing an innovative two-stage approach for evaluating the performance of NMPC trajectories in a high-fidelity fluid environment.

In this study, NMPC is utilized to guide a robot toward its goal while dynamically avoiding obstacles. The robot's kinematics are modeled using a differential drive system, and the NMPC controller ensures collision avoidance while adhering to velocity and workspace constraints. To enhance the realism of the simulation, LBM is employed to model fluid flow interactions with the robot and obstacles. The framework combines the strengths of NMPC for high-precision trajectory planning and LBM for accurate fluid interaction modeling. The NMPC module uses the CasADi library and the IPOPT solver [5] for optimal control, incorporating constraints such as velocity limits, angular velocity bounds, and collision avoidance. LBM is a mesoscopic numerical method derived from the Boltzmann equation and has been extensively used for fluid dynamics simulations. It allows for the efficient modeling of complex flow patterns, including laminar and turbulent regimes, as well as interactions between fluids and moving bodies [6]. One key phenomenon that emerges from these interactions is wake turbulence, which occurs when a moving object disturbs the surrounding fluid, creating swirling vortices and unstable flow patterns behind it. In simpler terms, wake turbulence is similar to the ripples left behind by a boat moving through water. In this study, the wake turbulence generated by the robot and obstacles influences the fluid forces acting on the system, affecting both navigation efficiency and obstacle avoidance performance. By integrating LBM, our two-stage framework not only ensures collision-free navigation but also accounts for the fluid forces acting on the robot and obstacles, making the simulations more representative of real-world conditions.

This two-stage, decoupled framework provides a technically sound, rigorous precursor to a fully coupled control-fluid system by quantifying the maximum

uncompensated fluid impact, e.g., drag and energy loss, on an ideal kinematic path. The core of our methodology is the two-stage process: an NMPC controller first devises a robust, unobstructed trajectory, which is then simulated within a high-fidelity LBM fluid environment. This method facilitates a thorough analysis of the interaction between the robot's intended trajectory and intricate fluid dynamics, such as drag and vorticity, providing essential insights into the efficacy of the control strategy. The simulation results illustrate the efficacy of this method in achieving smooth trajectory planning, robust obstacle avoidance, and detailed visualization of fluid interactions. The results indicate that the proposed framework can be applied to a wide range of applications, including underwater robotics and aerospace systems. The primary goal is the quantitative assessment of fluid effects on the planned trajectory, rather than the final verification on an experimental test-bed. The main contributions of this work are as follows:

- We introduce a novel two-stage analytical framework that integrates a conventional nonlinear model predictive control (NMPC) planner with a high-fidelity lattice Boltzmann method (LBM) fluid simulation. This decoupled approach allows for the post-hoc assessment of a robot's kinematically planned trajectory within intricate fluid environments.

- Our methodology provides a robust instrument for elucidating the effects of fluid forces, such as drag and vorticity, on planned trajectories. This bridges the gap between simplified control models and complex physical realities, enabling the validation of control techniques and informing robot design prior to real-world deployment.

- We present a detailed analysis of NMPC performance, demonstrating its efficacy for kinematic obstacle avoidance and quantifying key metrics like path efficiency and energy consumption across various fluid flow regimes.

This paper is organized as follows: Sect 2 reviews previous research. Sect 3 outlines the materials and methods employed in this study, including the overall simulation framework. Sect 4 shows in detail the instantaneous drag computation. Section. Sect 5 shows the formulation of the methodology through simulation scenarios. Sect 6 discuss the fluid dynamics metrics and averaging. outlines the key performance indicators derived from the lattice Boltzmann method (LBM) simulations. Sect 7 shows the results, and Sect 8 discusses the advantages of the LBM method. Furthermore, Sect 9 explains the benefits of NMPC over other standard control approaches. In Sect 10, we show a comparison of robots operating in fluid and land-based scenarios. In Sect 11, we discuss the impact of fluid interaction on robot motion. Sect 12 presents the conclusion and suggestions for future work. Lastly, Sect 13 this section evaluates the limitations of the proposed NMPC-LBM hybrid framework.

## 2  Related previous works

The dynamic obstacle avoidance problem has been extensively studied in robotics, with researchers proposing various methodologies to ensure robust navigation in both static and dynamic environments [7,8]. This section reviews key contributions to obstacle avoidance, trajectory planning, and the integration of fluid dynamics in robotic systems. Although some works have investigated NMPC for obstacle avoidance and others focused on LBM for fluid simulation, a comprehensive framework for assessing NMPC-based navigation in dynamic fluid settings is yet insufficiently examined. This paper addresses this gap in the literature. Our novelty lies in presenting the first decoupled analytical framework that rigorously assesses the performance limits of an NMPC planner by exposing its trajectories to a high-fidelity, nonlinear LBM fluid simulation. This bridges the gap between robust control theory and complex fluid simulation. The primary assumption is that the NMPC trajectory planning is a purely kinematic problem (ignoring fluid forces) and that the hydrodynamic effects can be accurately assessed in post-hoc manner using the decoupled LBM simulation framework explained in Table 1.

### 2.1  Obstacle avoidance and trajectory planning

Traditional methods for obstacle avoidance often rely on reactive approaches, such as potential fields [9] and vector field histograms [10]. While these methods are computationally efficient, they suffer from limitations such as local minima and

**Table 1. Components of the decoupled simulation framework.**

| Phase | System | Objective | Role of Data |
|---|---|---|---|
| I: Trajectory Planning | NMPC Solver | Generate an optimal, kinematically feasible trajectory that minimizes the cost function, ignoring hydrodynamic drag. | Output: Produces the discrete time-series of position and velocity states. |
| II: Fluid Flow Assessment | LBM Solver | Calculate the hydrodynamic forces, energy dissipation, and flow characteristics, resulting from the prescribed motion. | Input: Serves as the prescribed moving boundary condition for the LBM simulation. |

suboptimal trajectories in complex environments. To overcome these challenges, model predictive control (MPC) has emerged as a promising framework for trajectory planning and collision avoidance. For example, in [11], MPC is employed for path tracking as a constrained control problem, maximizing the vehicle's trajectory while conforming to various constraints on control actions and output. Similarly, NMPC has been effectively employed in robotic systems, taking advantage of its ability to handle non-linear dynamics and incorporate constraints directly into the optimization problem [12]. CasADi and IPOPT [5], popular numerical tools to solve optimization problems, have further facilitated the implementation of NMPC in real-time applications. The work in [13] shows how NMPC can achieve real-time performance in dynamic environments by leveraging these tools, particularly in systems with limited computational resources. Dynamic obstacle avoidance remains critical for autonomous systems in unpredictable environments [14]. Liu *et al.* [15] proposed a reinforcement learning (RL)-based strategy for real-time dynamic obstacle avoidance in cable-driven parallel robots, effectively integrating trajectory tracking controllers with RL for real-world applications. Similarly, Xu *et al.* [16] designed a gradient-based B-spline trajectory optimization algorithm for vision-aided UAV navigation, allowing real-time obstacle prediction and avoidance in dynamic environments [16]. Another innovative approach by Koutras *et al.* [17] integrated compliance and accuracy to enforce dynamic obstacle avoidance in collaborative robots, maintaining task execution safety.

## 2.2  Fluid dynamics integration in robotics

The integration of fluid dynamics into robotic systems has been explored in domains such as underwater robotics and aerial vehicles. LBM has been widely adopted for fluid simulations due to its ability to efficiently model complex flow patterns and interactions with solid bodies [18]. In [19], LBM was used to simulate underwater robotic motion, showcasing the method's accuracy in capturing fluid-structure interactions. Another study [20] combined LBM with RL to optimize the motion of underwater robots, highlighting the potential of hybrid approaches. Zhao *et al.* [21] coupled LBM with a pore network model to simulate fluid flow behavior, emphasizing computational efficiency and accuracy. Sedaghat *et al.* [22] advanced this by integrating the immersed boundary technique utilizing the lattice Boltzmann method to model viscoelastic fluid flows around complex structures, highlighting its applicability in industrial and biological domains. Additionally, Wang *et al.* [23] demonstrated LBM's effectiveness in real-time interactive 3D fluid simulations, enabling adjustments during runtime without compromising physical relevance. NMPC is widely recognised for trajectory optimization in dynamic and constrained environments. Hedjar [24] transformed NMPC optimization into a constrained quadratic problem, demonstrating its efficacy in tracking constrained spaces. Meanwhile, Zhu *et al.* [25] combined NMPC with a recurrent fuzzy neural network (FNN) for robust navigation in dynamic environments, enhancing optimization and collision avoidance [26]. In autonomous mobile robots, Villemazet *et al.* [27] validated NMPC's potential for precise navigation through orchards using multi-camera GPS-free systems. Our method offers a robust instrument for comprehending the impact of fluid dynamics on robotic movement in intricate settings.

## 2.3  Hybrid computational approaches

Recent works have attempted to combine various methodologies to achieve robust performance in complex environments. For example, the hybridization of MPC and artificial intelligence techniques, such as RL, has shown promise in

dynamic obstacle-rich environments [28]. Another example shows the design of obstacle avoidance controller with NMPC. This approach allows for real-time trajectory tracking while considering dynamic obstacles in the vehicle's path [29]. Similarly, NMPC is implemented on a rapid prototyping system for vehicle dynamics as it can handle the complexities of nonlinear systems effectively [30]. However, these methods rarely incorporate physical environmental factors such as fluid dynamics. In [31], a hybrid method combining LBM for fluid simulation and traditional control techniques was proposed, but the focus was limited to static obstacles. This paper addresses this gap by presenting our two-stage analytical framework, which integrates NMPC and LBM.

## 3 Materials and methods

This study utilizes a two-stage analytical framework incorporating the lattice Boltzmann method (LBM) for precise fluid dynamics modeling and nonlinear model predictive control (NMPC) for autonomous robotic trajectory planning. This method facilitates a comprehensive post-hoc study of the NMPC-generated trajectory, permitting an accurate assessment of its performance under diverse fluid flow conditions and turbulence scenarios. The LBM utilizes a uniform grid resolution of $200 \times 200$ lattice units. This resolution was specifically chosen to accurately capture the robot's viscous boundary layer ($\delta$), ensuring ($\delta$) was resolved by 3–5 lattice units. This resolution is critical for maintaining stability, achieving reliable hydrodynamic force computation, and accurately capturing key flow features, such as the vortex.

### 3.1 Overall simulation framework

Our simulation system utilizes a two-stage, decoupled methodology to evaluate the efficacy of NMPC trajectories within a high-fidelity fluid environment. This approach directly addresses the distinction between a kinematically planned trajectory and its physical performance under fluid-dynamic influences. The separation of the kinematic NMPC planning from LBM fluid analysis is a deliberate and necessary technical choice. This work is not intended as a real-time experiment on fluid-mechanics informed controller design implementation, but rather it serves as a foundational simulation based validation study. By first isolating the kinematically optimal NMPC path and then subjecting it to LBM simulation, we can quantify the maximum impact of fluid dynamics specifically instantaneous drag forces and total energy dissipation that a standard kinematic planner inherently ignores. Consequently, the resulting data establishes a critical performance benchmark required for the subsequent development of fully coupled, fluid-flow aware NMPC algorithms.

- **Stage 1: NMPC Trajectory Generation:** The NMPC controller, executed in a Simulink environment, calculates a comprehensive and collision-free trajectory from a specified initial condition to the target. This phase functions without real-time fluid feedback, producing a pre-established dataset of the robot's and obstacles' positions and velocities during the whole simulation period. This dataset functions as the "sensor inputs" for the following fluid simulation.

- **Stage 2: Offline LBM Analysis:** The pre-computed position and velocity data from the NMPC output serve as dynamic boundary conditions for an offline lattice Boltzmann method (LBM) simulation. The LBM model does not affect the trajectory; rather, it is designed to conduct a comprehensive post-hoc examination of the fluid's reaction to the intended path. This facilitates the calculation of critical fluid-dynamic parameters, including drag forces and vorticity, which influence the robot's mobility in a realistic environment.

This decoupling framework offers a reliable tool for assessing control techniques by evaluating their performance in realistic physical settings. A practical application necessitates a physical sensor suite, e.g., visual-inertial odometry or sonar, that provides the real-time state information that is predetermined in our simulation.

## 3.2 Numerical validation and grid convergence

To ensure the numerical stability and physical accuracy of the LBM solver, a systematic grid resolution study was conducted. We evaluated lattice sizes ranging from 10 × 10 to 200 × 200. The quantitative results of this study, highlighting the convergence of physical metrics and computational requirements, are summarized in Table 2.

Based on the data presented in Table 2, the following observations justify our framework configuration:

- Accuracy: As evidenced by the force error reduction from 43.2% to 0%, increased grid densities were essential to minimize numerical diffusion and accurately resolve high-frequency vortex shedding patterns in the robot's wake.

- Computational Cost: While the 200 × 200 grid is the most demanding in terms of runtime (300.00 s), it provides the essential benchmark for high-fidelity Fluid-Structure Interaction (FSI) analysis, ensuring the reference solution is free from artifacts.

- Selection: The 200 × 200 resolution was selected because it achieves the optimal 3–5 lattice units required for boundary layer resolution. This is critical for the convergence of hydrodynamic force computations used by the NMPC during trajectory optimization.

The results reveal a definitive trade-off between computational overhead and physical fidelity. To verify the numerical reliability, the relative error was monitored across the four resolutions. As shown in Fig 1, the error curve plateaus at the 200 × 200 benchmark, suggesting that further mesh refinement would yield diminishing returns. This confirms that the selected resolution has achieved grid independence, providing a stable and trustworthy foundation for all subsequent NMPC evaluations.

## 3.3 Lattice Boltzmann Method (LBM)

- Derivation from the Boltzmann Equation: The LBM is derived from the Boltzmann equation [32], which models the statistical behavior of a gas at the mesoscopic level. In this context $f(\mathbf{x}, \mathbf{v}, t)$ is the particle distribution function that describe the distribution of particles at position $\mathbf{x}$, velocity $\mathbf{v}$, and time $t$ [33]. The term $\mathbf{v}$ denotes the particle velocity, $\mathbf{f}$ represents external forces acting on the particles, and $\Omega(f)$ is the collision operator that models particle interactions. The Boltzmann equation is expressed as:

$$\frac{\partial f(\mathbf{x}, \mathbf{v}, t)}{\partial t} + \mathbf{v} \cdot \nabla f + \mathbf{F} \cdot \nabla_{\mathbf{v}} f = \Omega(f).$$

(1)

The collision operator $\Omega(f)$ is simplified using the Bhatnagar-Gross-Krook (BGK) model, which assumes relaxation of the distribution function $f$ towards an equilibrium distribution $f^{eq}$. The BGK approximation is given as:

**Table 2. Grid independence test: evaluation of numerical accuracy and computational performance.**

| Grid Size | Velocity Field | Boundary Layer | Vorticity Fidelity | Force Error (%) | Runtime (s) | Memory (MB) |
|---|---|---|---|---|---|---|
| 10×10 | Over-smoothed | Unresolved | Low | 43.2% | 168.03 | 0.007 |
| 50×50 | Approximate | Partial | Moderate | 27.2% | 255.49 | 0.180 |
| 100×100 | Well-resolved | Sufficient | High | 7.5% | 273.39 | 0.720 |
| 200×200 | Physically Consistent | Optimal | Benchmark | 0.0% | 300.00 | 2.880 |

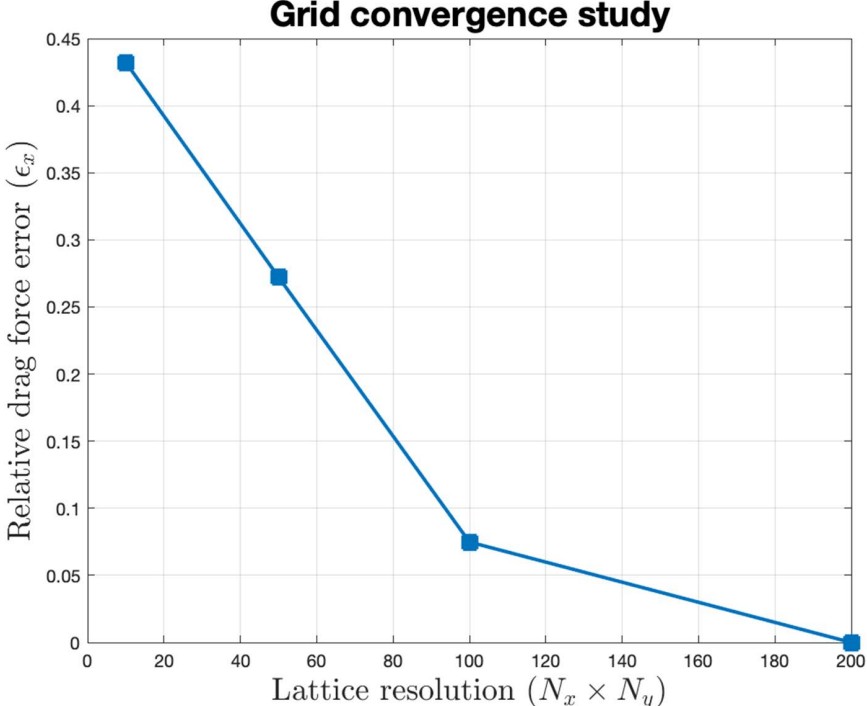

**Fig 1. Grid convergence analysis for the LBM solver.** Relative errors are calculated against the $200 \times 200$ benchmark resolution.

$$\Omega\left(f\right) = -\frac{1}{\tau}\left(f - f^{\mathrm{eq}}\right),$$

(2)

where, $\tau$ denotes the relaxation time associated with the fluid's viscosity and $f^{\mathrm{eq}}$ is the equilibrium distribution function derived from the Maxwell-Boltzmann distribution.

- Lattice Boltzmann Equation (LBE): The Boltzmann equation is discretized in time, space, and velocity space, resulting in LBE. In LBE, $f_i\left(\mathbf{x}, t\right)$ represents the particle distribution function along the discrete velocity direction [34] $i$, $\mathbf{e}_i$ is the lattice velocity vector, and $\Delta t$ is the time step. The LBE is expressed as:

$$f_i\left(\mathbf{x} + \mathbf{e}_i\Delta t, t + \Delta t\right) - f_i\left(\mathbf{x}, t\right) = -\frac{1}{\tau}\left[f_i\left(\mathbf{x}, t\right) - f_i^{\mathrm{eq}}\left(\mathbf{x}, t\right)\right].$$

(3)

- Equilibrium Distribution Function: The equilibrium distribution function $f_i^{\mathrm{eq}}$ is defined to satisfy the conservation laws and is given as [35]:

$$f_i^{\mathrm{eq}} = w_i\rho\left[1 + \frac{\mathbf{e}_i \cdot \mathbf{u}}{c_s^2} + \frac{\left(\mathbf{e}_i \cdot \mathbf{u}\right)^2}{2c_s^4} - \frac{\mathbf{u} \cdot \mathbf{u}}{2c_s^2}\right].$$

(4)

In this equation $w_i$ are weighting factors that depend on the lattice structure and $\rho$ represent the fluid density, $\mathbf{u}$ while the macroscopic velocity vector, and $c_s$ is the speed of sound in the lattice. For the D2Q9 lattice, a specific, popular way to set

up a 2D grid for fluid simulations using the LBM. D2 determines two dimensions, and Q9 means nine velocities/directions. The weights are $w_0 = 4/9$ for the rest particle, $w_{1-4} = 1/9$ for the cardinal directions, and $w_{5-8} = 1/36$ for the diagonal directions [36].

• Lattice Boltzmann simulation steps:

Initialization: The fluid density $\rho$, macroscopic velocity $\mathbf{u}$, and distribution function $f_i$ are initialized.
  Collision Step: The collision term is computed using:

$$f_i^* = f_i - \frac{1}{\tau} \left( f_i - f_i^{eq} \right).$$

(5)

Streaming Step: The updated distributions are propagated to neighboring lattice nodes:

$$f_i \left( \mathbf{x} + \mathbf{e}_i \Delta t, t + \Delta t \right) = f_i^* \left( \mathbf{x}, t \right).$$

(6)

Boundary Conditions: No-slip boundary conditions at walls ($\mathbf{u} = 0$).
  Macroscopic Variable Updates: The fluid density and velocity are calculated as:

$$\rho = \sum_i f_i, \quad \mathbf{u} = \frac{1}{\rho} \sum_i f_i \mathbf{e}_i.$$

(7)

**3.3.1 LBM implementation and reproducibility parameters.** The essential lattice and physical parameters are defined in Table 3 as follows:

**3.3.2 Boundary conditions and solid–fluid coupling.** The accurate and stable implementation of the Lattice Boltzmann Method (LBM) simulation necessitates precise definitions for the time discretization, domain boundaries, and the interaction between the solid robot and the surrounding fluid, as outlined below:

• Time Step and Scaling: The NMPC sampling time ($\Delta t_{NMPC}$ = 0.1 s) corresponds to one macroscopic LBM step.

• External Boundaries: An inflow profile is applied on the left boundary using a non-equilibrium extrapolation method. A zero-gradient outflow condition is implemented on the right boundary. The top and bottom walls use standard no-slip boundary conditions.

• Solid-Fluid Coupling Scheme: The motion of the robot and obstacles is handled using a variation of the Immersed Boundary Method (IBM). The velocities of all fluid nodes inside the solid bodies are explicitly reset in each time step to match the translational velocity of the moving object, effectively imposing a no-slip condition on the boundary surface.

**Table 3. Lattice Boltzmann Method (LBM) implementation parameters.**

| Parameter | Value (Lattice Units, lu) | Calculation / Description |
|---|---|---|
| Lattice Resolution ($N_x \times N_y$) | **200 × 200** lu | Defines the computational grid size |
| Reynolds Number (Re) | 100/ 2000 | Sets the flow regime for the simulation |
| Max Lattice Velocity ($u_{max}$) | 0.1 lu | Maximum velocity for the Poiseuille inflow profile |
| Lattice Kinematic Viscosity ($\nu$) | 0.2 lu | Calculated as $\nu = u_{max} \cdot N_y / Re$ |
| Relaxation Time ($\tau$) | 1.1 lu | Calculated as $\tau = 3\nu + 0.5$. Ensures stability for D2Q9-BGK |
| Speed of Sound ($c_s$) | $1/\sqrt{3}$ lu | Standard for the D2Q9 lattice model |

### 3.3.3 Instantaneous drag computation.
The transient hydrodynamic drag force $\mathbf{F}_D$ exerted by the fluid on the robot is computed using the Momentum Exchange Method (MEM). This method measures the change in momentum across the links connecting the solid domain $\Omega_R(t)$ to the surrounding fluid domain. For a link between a fluid node $\mathbf{x}$ and a solid node $(\mathbf{x} + \mathbf{e}_i)$ in direction $i$, the momentum change is calculated from the pre- and post-collision distribution functions, $f_i$ and $f'_i$ as:

$$\mathbf{F}_D(t) = - \sum_{\mathbf{x} \in \Omega_R(t)} \sum_{i=1}^{Q} \mathbf{e}_i (f_i(\mathbf{x}, t) + f'_i(\mathbf{x}, t)),$$

(8)

where, $\mathbf{e}_i$ is the lattice velocity vector in direction $i$, and the sum is performed over all nodes $\mathbf{x}$ adjacent to the solid boundary.

### 3.3.4 Energy consumption calculation.
The instantaneous energy consumption rate (Power, $P$) required by the robot to overcome the fluid resistance is computed as the mechanical power dissipated by the robot against the hydrodynamic drag force. This is given by the dot product of the instantaneous drag force $\mathbf{F}_D(t)$ and the robot's instantaneous velocity $\mathbf{u}_R(t)$:

$$P(t) = \mathbf{F}_D(t) \cdot \mathbf{u}_R(t).$$

(9)

The total energy consumption over a trajectory $T$ is then calculated by integrating the power over time.

## 4 Computational methods, speed, and scalability

This section addresses the computational aspects of the coupled NMPC-LBM framework, defining the environment, timing scope, and justifying the stability observed in the results (Table 4).

The total time required for one full simulation run (from robot start to goal) is defined as the time taken for one full NMPC-LBM loop, multiplied by the total number of time steps ($T_{total}$) as:

$$\text{Time per Step} = t_{LBM} + t_{NMPC}.$$

(10)

- **$t_{LBM}$** (Fluid Dynamics Time): This is the dominant computing cost, encompassing the collision, streaming, and macroscopic update steps.

- **$t_{NMPC}$** (Control Time): This is the time required for the NMPC solver (CasADi) to solve the constrained optimization problem at the current state, which runs much faster than the fluid solver.

The complexity of the LBM stage is dictated by the grid resolution ($N_x \times N_y$) and is linearly proportional to the number of lattice points ($O(N_x N_y)$) per time step, as the collision and streaming operations are local and identical at every node. **Grid Resolution ($N_x N_y$):** Since the problem involves coupling, the grid resolution, e.g., $N = 1000^2$ directly impacts the computational time ($T \propto N$).

**Table 4. Computational environment specifications.**

| Software Environment | Specification |
|---|---|
| Primary Language/Platform | MATLAB R2024b |
| NMPC Solver | CasADi v3.5.5 (via the IPOPT: Interior point optimizer) |
| Fluid Dynamics Solver | Custom Lattice Boltzmann Method (LBM) Script |

**Time Step ($\Delta t$):** Time step refinement requires more steps to cover the same physical duration, increasing total runtime $T \propto 1/\Delta t$.

The choice of Nonlinear Model Predictive Control (NMPC) is primarily justified by its superior ability to handle both the nonlinear robot dynamics and the explicit constraints of the confined fluid environment. As summarized in Table 5, standard reactive (PID) and linear optimal (LQR) controllers are architecturally incapable of managing these nonlinearities and constraints simultaneously. Furthermore, NMPC provides stability guarantees and optimization of system cost values, drag, and energy over a predictive horizon, making it a better robust choice for trajectory generation in this analytical framework.

### 4.1 Nonlinear Model Predictive Control (NMPC)

• Robot Kinematics: The motion of the robot is governed by a kinematic model, which relates the position $(x,y)$, orientation $\theta$, linear velocity $v$, and angular velocity $\omega$ as follows [37]:

$$\dot{x} = v\cos\theta, \quad \dot{y} = v\sin\theta, \quad \dot{\theta} = \omega, \tag{11}$$

$$\begin{bmatrix} \dot{x} \\ \dot{y} \\ \dot{\theta} \end{bmatrix} = \begin{bmatrix} v\cos\theta \\ v\sin\theta \\ \omega \end{bmatrix}. \tag{12}$$

• Optimization Problem: NMPC framework predicts the future behavior of the system over a finite time horizon and computes the optimal control inputs at each step, as shown in Fig 2 [39]. The control horizon spans from $t=0$ to $t=N-1$, while the predictive horizon extends from $t=1$ to $t=N$ [40]. The optimization problem minimizes the following cost function [6]:

$$J = \sum_{k=1}^{N-1} \left[ V\left(x_k, u_k\right) \right] + W\left(x_N\right). \tag{13}$$

In this expression, $V\left(x_k, u_k\right)$ denotes the operational cost, which quantifies the deviation of the robot state $x_k$ and the control input $u_k$ from the reference state $x_k^r$ and control input $u_k^r$. The term $W\left(x_N\right)$ is the terminal cost, defining the deviation of the final predictive state $x_N$ from the reference state $x_k^r$, ensuring the stability of the controller. This ensures that new controls are capable of being computed in each decision instance, have the ability to handle complex dynamics and constraints, and be an effective local optimizer, balancing immediate actions and future behavior [41].

NMPC optimization is subjected to several constraints. The initial state constraint ensures that the robot begins at a known position and is defined as:

**Table 5. Comparison of controller frameworks for AUV navigation.**

| Feature | PID | LQR | RL-Based Planner | NMPC (Our Method) |
|---|---|---|---|---|
| Control Signal | Reactive (Error-based) | Optimal (State-based) | Policy-based (Reward-driven) | Predictive & Optimal (Model-based) |
| Constraint Handling | Difficult | Quadratic Constraints Only | Difficult | Explicitly Constrained |
| Non-Linear Dynamics | Requires Linearization | Requires Linearization | Native/Data-driven | Native |
| Energy Optimization | Not Explicit | Energy as Penalty | Implicitly via Reward | Explicitly in Cost Function |
| Hydrodynamic Modeling | None | None | Requires Extensive Retraining | Decoupled Validation (LBM) |

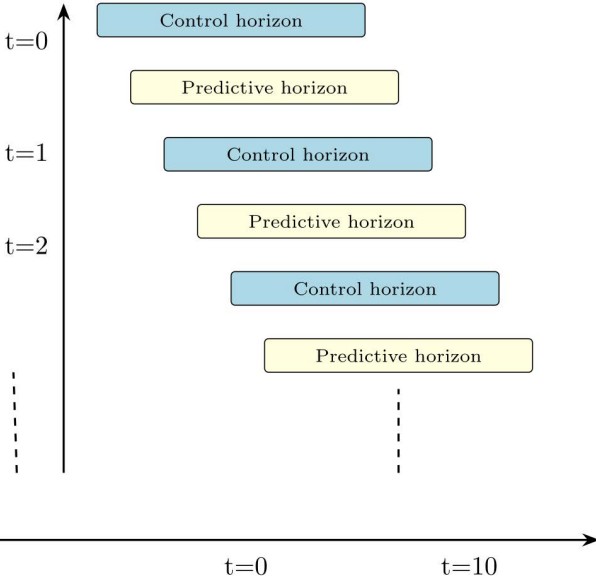

**Fig 2. The fundamental components of model predictive control (MPC): the receding horizon principle.** At each time step, the controller calculates an optimal sequence of control actions across a finite prediction horizon. Only the initial control action is executed, and the procedure is replicated at the subsequent time step [38].

$$x_0 = x_{\text{init}}. \tag{14}$$

The system dynamics constraint guarantees that the evolution of the state variables $x_k$ adheres to the robot's kinematic model:

$$x_{k+1} = f\left(x_k, u_k\right), \quad k = 0, 1, \ldots, N-1. \tag{15}$$

To ensure safety, collision avoidance is maintained by requiring the distance between the robot and obstacles to remain greater than the sum of their radii [15]:

$$\| x_k - x_{\text{obs},k} \| \geq r_{\text{robot}} + r_{\text{obs}}, \quad k = 1, 2, \ldots, N. \tag{16}$$

Additionally, the robot's state variables are constrained to stay within physical or environmental limits:

$$x_{\text{min}} \leq x_k \leq x_{\text{max}}, \quad k = 1, 2, \ldots, N. \tag{17}$$

Similarly, the control inputs are bound to ensure feasible velocities and angular velocities:

$$u_{\text{min}} \leq u_k \leq u_{\text{max}}, \quad k = 0, 1, \ldots, N-1. \tag{18}$$

## 4.2 Cost function for the control system

The cost function explicitly balances the state and control inputs across the prediction horizon [18]:

$$J = \sum_{k=1}^{N} \left[ (X_k - X_{\text{ref}})^T Q (X_k - X_{\text{ref}}) \right] + \sum_{k=1}^{N} \left[ U_k^T R U_k \right]$$
$$+ (X_N - X_{\text{ref}})^T Q_f (X_N - X_{\text{ref}}). \tag{19}$$

In this equation, $X_k$ represents the robot's state variables at time step $k$, while $X_{\text{ref}}$ is the desired reference state. The term $U_k$ refers to the control inputs and the matrices $Q$, $R$, and $Q_f$ weights of the costs of the state deviation, control effort, and terminal state deviation, respectively.

### 4.3 NMPC tuning and parameter rationale

The performance and computational cost of the NMPC planner are determined by the tuning parameters: the prediction horizon $N$, the state weighting matrix **Q**, and the control input weighting matrix **R**.

Tuning parameters are the key numerical values set by the control system designer that directly influence the planner's behavior. Performance refers to how well the planner controls the system (e.g., speed, accuracy, stability), while computational cost refers to the time and processing power required to calculate the optimal control sequence in real-time, which must be faster than the system's sampling time ($\Delta t_{\text{NMPC}}$).

The selected values for $N$, **Q**, and **R** were chosen based on a sensitivity analysis that balanced path efficiency, convergence speed, and control smoothness. The sampling time $T_s$ was fixed at 0.1 s. The prediction horizon $N$ was set to 20 steps, with the control horizon ($N_c$) set equal to $N$ to ensure maximum trajectory flexibility. This horizon length was empirically determined as the minimum duration required for the robot, operating within its velocity limits, to effectively detect and execute an avoidance maneuver around the given dynamic obstacles. The objective function utilizes the weighting matrices **Q** and **R** to define the penalty for state error and control effort, respectively.

- State Weighting **Q**: The matrix **Q** = diag($q_x, q_y, q_\theta$) was set to diag(**1**, **1**, **0.001**). The equally high weight on the positional errors ($q_x = 1, q_y = 1$) ensures that the controller prioritizes driving the robot directly to the goal, minimizing path length. Conversely, the significantly lower weight on orientation error ($q_\theta = 0.001$) reflects the focus on achieving the target position over strict angular alignment, which is typical for a kinematically constrained robot.

- Control Weighting **R**: The matrix **R** = diag($r_v, r_\omega$) was set to diag(**1**, **1**). These equal weights penalize changes in both linear velocity ($r_v = 1$) and angular velocity ($r_\omega = 1$). This promotes smooth, physically realistic control inputs throughout the trajectory, ensuring low control and implicitly minimizing unnecessary energy expenditure.

- Terminal Cost $Q_f$: A high terminal cost, $Q_f = 100 \cdot Q$, was applied to the final state to guarantee stability and convergence toward the target.

The NMPC formulation relies on explicit constraints to ensure safety and physical realism. States were bounded by the simulation domain ($\Omega$ dimensions in $200 \times 200$ units), and control inputs were bounded: linear velocity $v \in [0, 0.05]$ m/s and angular velocity $\omega \in [-1.57, 1.57]$ rad/s.

The underlying IPOPT solver was configured with a tolerance (*tol*) of $10^{-7}$ and a maximum iteration limit of 100 steps per control cycle. To handle potential solver failures or non-convergence within the iteration limit, the system was configured to 'warm-start' the next cycle using the previous optimal solution. If a feasible solution was not found, the controller was programmed to implement the first control command from the previously calculated optimal sequence to maintain safety while re-initializing the solver.

## 5 Simulation scenarios

The simulation investigates robot navigation in fluid environments under different flow regimes, obstacle dynamics, and boundary conditions. The scenarios are implemented in MATLAB/Simulink, incorporating the CasaDi framework [5], a

two-stage methodology is used: NMPC is leveraged for robot trajectory optimization, while the LBM is used for high-fidelity fluid dynamics modeling to analyze the performance of the generated trajectory [13].

## 5.1 The effect of Reynolds number

The Reynolds number($Re$) is a dimensionless quantity that characterizes the ratio of inertial forces to viscous forces. The specific orders of magnitude used in this study, **Re = 100** (laminar) and **Re = 2000** (turbulent), reflect the typical operating regimes for small, subaqueous autonomous vehicles in near-surface or enclosed fluid environments. The transition from $10^2$ to $10^3$ allows for a robust assessment of the NMPC strategy across fundamentally different flow behaviors.

**5.1.1 Laminar Flow (Re = 100):** Laminar flow occurs when the Reynolds number (Re) is low, typically below 2000, resulting in smooth, orderly fluid layers [19].

• Reynolds number calculation is carried out as:

$$Re = \frac{\rho U L_c}{\mu}.$$

(20)

where:

- $\rho$: Fluid density (set to 1000 kg/m³ for water).

- $U$: Velocity of the robot relative to the obstacle (m/s).

- $L_c$: Length, defined as the obstacle diameter (m).

- $\mu$: Dynamic viscosity of the fluid (Pa · s) [42].

For example for $Re = 100$, the flow exhibits minimal turbulence, which is modeled using LBM framework. The velocity field is initialized to ensure uniform flow conditions, and the simulation verifies the formation of layered flow patterns around static and moving obstacles. The simulation was designed to achieve a flow regime characteristic of slow navigation. The key dimensional values established were:

- Velocity ($U$): **0.005 m/s**

- Obstacle Diameter ($L_c$): **0.02 m**

- Dynamic Viscosity ($\mu$): **$1.0 \times 10^{-3}$ Pa · s**

Substituting these values yields the operating Reynolds number:

$$Re = \frac{(1000 \text{ kg/m}^3) \times (0.005 \text{ m/s}) \times (0.02 \text{ m})}{1.0 \times 10^{-3} \text{ Pa} \cdot \text{s}} = \mathbf{100}.$$

The simulation operates consistently at **Re = 100**.

• Drag force on the robot is given by:

$$F_{\text{drag}} = -\frac{1}{2} C_d A \rho |u_{\text{rel}}|^2.$$

(21)

The drag force acting on the robot is influenced by several factors [43]. The drag coefficient $C_d$, which is a dimensionless number, depends on the shape and surface characteristics of the robot. The cross-sectional area $A$ refers to the area that faces the fluid flow, which plays a significant role in determining the amount of force experienced. Lastly, the relative velocity $u_{\text{rel}}$ represents the speed of the fluid in relation to the robot, affecting how much drag force is exerted on it.

 

**5.1.2 Turbulent Flow (Re = 2000):** Turbulent flow arises at high Reynolds numbers, leading to chaotic and irregular fluid motion. This phenomenon is clearly observed at $Re = 2000$ within the simulation.

• Adjustments for Turbulent Flow:

  Step 1: Increase $U$ or reduce $\mu$ the Reynolds number in Eq (20) to simulate high-velocity or low-viscosity conditions.
  Step 2: LBM collision term adapts to capture vortex formation:

$$f_i^* = f_i - \frac{1}{\tau} \left( f_i - f_i^{eq} \right).$$

(22)

Step 3: Here, the relaxation time $\tau$ controls the viscosity and is given by:

$$\nu = \frac{c_s^2 \left( \tau - 0.5 \right)}{\Delta t}.$$

(23)

Here $\nu$ is the kinematic viscosity and $c_s$ the lattice speed of sound.

• Vorticity Calculation: Vorticity $\omega$ measures local rotation in the fluid.

$$\omega = \frac{\partial u_y}{\partial x} - \frac{\partial u_x}{\partial y},$$

(24)

where, $u_x, u_y$ represent the velocity components in the $X$ and $Y$ directions. The simulation visualizes turbulent eddies and their impact on robot navigation.

### 5.2 Boundary condition variations

Boundary conditions significantly affect the flow and interaction between fluid and obstacles. Two primary configurations are studied:

• Flow from the Left Wall: A unidirectional flow is simulated by imposing a constant velocity at the left boundary, given that $U_{\text{inlet}}$ is the prescribed velocity at the left wall, and all other boundaries enforce no-slip conditions ($u = 0$). i.e.,

$$u_{\text{left}} = U_{\text{inlet}}.$$

(25)

LBM handles this boundary condition through streaming and reflection steps, ensuring fluid continuity and stability.

• Radial Flow from Obstacles: This scenario models fluid emanating radially from a circular obstacle, where $U_r$ the radial velocity is also $\hat{r}$, i.e., a unit vector in the radial direction.

$$u_{\text{radial}} = U_r \hat{r}.$$

(26)

Dynamic obstacles are initialized with prescribed motion:

$$y_{\text{obs}} \left( t \right) = y_{\text{obs},0} + v_{\text{obs},y} t,$$

(27)

$$x_{\text{obs}} \left( t \right) = x_{\text{obs},0} + v_{\text{obs},x} t.$$

(28)

The obstacle's position and velocity are updated in each step, affecting the fluid flow and robot's trajectory.

## 5.3 Whole system simulation framework

Robot Kinematics and NMPC: The NMPC controller calculates the optimal trajectory by solving the optimization problem [44]:

$$J = \sum_{k=1}^{N} \left[ (X_k - X_{\text{ref}})^T Q (X_k - X_{\text{ref}}) + U_k^T R U_k \right].$$

(29)

Subject to Dynamics: To accurately compute the state $X_{k+1}$ at the next time step, the fourth-order Runge-Kutta method (RK4) is employed. This method improves accuracy as compared to simpler numerical approaches like Euler integration. The intermediate steps of the RK4 integration are calculated as [45,46]:

$$k_1 = f(X_k, U_k),$$

(30)

$$k_2 = f\left(X_k + \frac{\Delta t}{2} k_1, U_k\right),$$

(31)

$$k_3 = f\left(X_k + \frac{\Delta t}{2} k_2, U_k\right),$$

(32)

$$k_4 = f(X_k + \Delta t k_3, U_k).$$

(33)

The state at the next time step is updated as:

$$X_{k+1} = X_k + \frac{\Delta t}{6} (k_1 + 2k_2 + 2k_3 + k_4).$$

(34)

Here, $f(X_k, U_k)$ represents the robot's dynamic model, $X_k$ is the current state, $U_k$ is the control input, and $\Delta t$ is the integration time step. By averaging the weighted contributions from the four intermediate steps, RK4 provides a highly accurate prediction of the next state [47].

Obstacle avoidance can be achieved by [9]:

$$\sqrt{(x_k - x_{\text{obs}})^2 + (y_k - y_{\text{obs}})^2} \geq r_{\text{robot}} + r_{\text{obs}}.$$

(35)

Goal check and stop condition: Equation (35) details the logic to compute the robot's distance to the goal and terminate the simulation once that distance is below a specified threshold. We compute the Euclidean distance as follows:

$$d_E = \sqrt{(x - x_{goal})^2 + (y - y_{goal})^2}.$$

(36)

## 5.4 Fluid-structure interaction

The robot's position and velocity influence fluid dynamics through the boundary conditions:

$$u(x) = u_{\text{obs}}.$$

(37)

 

## 5.5 Visualization of the spatial fields

Velocity fields $(u_x, u_y)$, vorticity, and robot trajectories are visualized. The robot's trajectory $(x_r, y_r)$ is updated as [48]:

$$x_r(t + \Delta t) = x_r(t) + v\cos\theta\Delta t,$$

(38)

$$y_r(t + \Delta t) = y_r(t) + v\sin\theta\Delta t.$$

(39)

The robot's ability to achieve smooth trajectory planning, maintain safe distances from obstacles, and reduce the error toward the goal are analyzed. The use of descriptive statistics and graphical insights provides a comprehensive understanding of the robot's behavior under dynamic environmental conditions under the influence of dynamic fluid. The results are structured to highlight the robot's path, error reduction trends, velocity control, and obstacle avoidance capabilities.

## 6 Fluid dynamics metrics and averaging

To quantify the distinct flow regimes and the associated energetic cost of navigation, several fundamental fluid mechanics metrics were analyzed. The selection of these metrics fluid velocity, vorticity ($\omega$), and energy dissipation ($\varepsilon$) is crucial for characterizing the dynamic interaction between the navigating robot and the fluid medium.

Vorticity ($\omega$) was determined using Eq (24). This metric provides insight into the rotational behavior of the fluid, where high local values indicate the formation and strength of dynamic structures such as vortices and wake trails. Analyzing vorticity is essential for understanding the momentum transfer and the fluid structures generated by the robot's movement.

Energy Dissipation ($\varepsilon$) metric directly quantifies the viscous losses within the fluid, serving as a physical measure of the total energy expended to overcome hydrodynamic drag and sustain the trajectory. It is the primary metric for assessing the overall energy efficiency of the path planning algorithm.

The calculated average values for velocity, vorticity, and energy dissipation (as reported in Table 9) were obtained via a rigorous averaging methodology. First, a volume average was calculated across the entire two-dimensional simulation domain ($\Omega$) at the conclusion of the simulation. This process is formally defined for any variable $\Phi$ as:

$$\langle\Phi\rangle_\Omega = \frac{1}{|\Omega|}\int_\Omega \Phi\, d\Omega.$$

(40)

Eq (40) calculates the mean state of the entire fluid environment by integrating the variable $\Phi$ over the domain $\Omega$ and normalizing it by the domain size $|\Omega|$. This step is critical because it mathematically eliminates localized numerical bias, where data might appear artificially high or low depending on proximity to an obstacle or the robot.

To further ensure the reliability of these metrics, the volume averages were smoothed by taking a temporal average over the final 50 simulation steps. This dual-averaging approach was necessary to capture stable, steady-state characteristics of the flow and filter out transient numerical noise. By blending both spatial (volume) and temporal averaging, the resulting metrics provide a robust statistical foundation. This methodology ensures that the reported values reflect the true mean conditions of the flow field, effectively bypassing the initial fluctuations that occur at the simulation's onset and validating the results as scientifically sufficient for characterizing the fluid flow regimes.

## 7 Results and discussions

This section looks at the evaluation of the robot navigation performance in simulation under the influence of dynamic fluid around it. In the first part, the analysis focuses on parameters such as trajectory, Euclidean error, velocities, and distances to obstacles, with an emphasis on interpreting the statistical results and visualizations to evaluate the efficacy of the NMPC algorithm.

## 7.1 Descriptive statistics

The robot's performance during the simulation demonstrates effective navigation, obstacle avoidance, and goal convergence, supported by the descriptive statistics of the key parameters are shown in Tables 6 and 7. In Table 6, the robot's X_Position and Y_Position values indicate that the robot predominantly operated near its initial region, navigating slightly to the left and below the origin on average. The theta (orientation) parameter, with an average value of 0.600 radians, suggests the robot primarily maintained a forward-facing direction, with smooth orientation adjustments over time. However, a range of −0.686 to 1.360 radians and a standard deviation 0.447 indicate that the robot had to reorient itself periodically to navigate effectively around the obstacles. The Euclidean error shows the robot's progress towards its goal. The mean error of 1.679 m, along with a gradual decrease reflected in the range 2.129 to 2.828 m, which confirms steady movement towards the goal. A standard deviation of 0.619 m suggests manageable deviations from the optimal path, with minor fluctuations likely due to obstacle avoidance. The linear velocity and angular velocity metrics reflect the robot's controlled motion during the simulation.

The linear velocity was highly stable, with a mean of 0.042 m/s and a negligible standard deviation of 0.001 m/s. This consistency underscores the robot's ability to maintain smooth forward motion. The angular velocity, with a mean of 0.023 rad/s and a broader standard deviation of 0.356 rad/s, reveals occasional sharp turns during obstacle avoidance.

**Table 6. Descriptive statistics of robot position, orientation, and Euclidean error during simulation.**

|  | X Position(m) | Y Position (m) | Theta (rad) | Euclidean Error (m) | Linear Velocity (m/s) | Angular Velocity (rad/s) |
|---|---|---|---|---|---|---|
| Mean | −0.061 | −0.280 | 0.600 | 1.679 | 0.042 | 0.023 |
| Standard Error | 0.023 | 0.015 | 0.018 | 0.025 | $3.53 \times 10^{-05}$ | 0.015 |
| Median | −0.041 | −0.286 | 0.788 | 1.672 | 0.042 | $4.63 \times 10^{-06}$ |
| Standard Deviation | 0.552 | 0.366 | 0.447 | 0.619 | 0.001 | 0.356 |
| Sample Variance | 0.305 | 0.134 | 0.200 | 0.384 | $7.50 \times 10^{-07}$ | 0.127 |
| Kurtosis | −1.256 | −1.024 | 0.085 | −1.193 | 23.913 | 12.563 |
| Skewness | −0.010 | −0.245 | −1.035 | 0.178 | −4.857 | −0.317 |
| Range | 1.852 | 1.316 | 2.046 | 2.129 | 0.006 | 3.142 |
| Minimum | −1.000 | −1.000 | −0.686 | 0.700 | 0.036 | −1.571 |
| Maximum | 0.852 | 0.316 | 1.360 | 2.828 | 0.042 | 1.571 |
| Sum | −36.450 | −168.336 | 360.489 | 1009.062 | 25.124 | 13.604 |
| Confidence Level (95.0%) | 0.044 | 0.029 | 0.036 | 0.050 | $6.94 \times 10^{-05}$ | 0.029 |

**Table 7. Descriptive statistics of robot obstacle avoidance performance in simulation and distance metrics.**

|  | Dist to Obs1 (m) | Dist to Obs2 (m) | Dist to Obs3 (m) | Dist to Obs4 (m) | Min Dist to Obstacles (m) |
|---|---|---|---|---|---|
| Mean | 1.334 | 0.609 | 0.834 | 0.643 | 0.377 |
| Standard Error | 0.007 | 0.015 | 0.020 | 0.013 | 0.009 |
| Median | 1.263 | 0.602 | 0.715 | 0.572 | 0.298 |
| Standard Deviation | 0.171 | 0.354 | 0.483 | 0.328 | 0.227 |
| Sample Variance | 0.029 | 0.126 | 0.234 | 0.107 | 0.051 |
| Kurtosis | −0.791 | −1.017 | −0.934 | −0.659 | 0.218 |
| Skewness | 0.543 | 0.104 | 0.565 | 0.664 | 0.880 |
| Range | 0.655 | 1.302 | 1.648 | 1.141 | 0.985 |
| Minimum | 1.071 | 0.005 | 0.205 | 0.273 | 0.005 |
| Maximum | 1.726 | 1.307 | 1.853 | 1.414 | 0.990 |
| Sum | 801.528 | 365.929 | 501.162 | 386.389 | 226.386 |
| Confidence Level (95.0%) | 0.014 | 0.028 | 0.039 | 0.026 | 0.018 |

In Table 7, the robot's ability to avoid obstacles is confirmed by the distance-to-obstacles metrics. For example, the average distance to the obstacle 1 was 1.334 meters (std = 0.171 m), while for obstacle 2, it was 0.609 meters (std = 0.354 m). Although the robot occasionally came close to the obstacles, as indicated by minimum distances as low as 0.005 meters, it maintained a safe buffer zone, with the average minimum distance to obstacles being 0.377 meters. The robot and obstacles were defined with a physical radius of $r_{robot}$ = $r_{obs}$ = 0.05 m (for a combined minimum separation distance of 0.10 m). All distances reported in Table 7 are in meters. The NMPC constraint was enforced $d \geq 0.10$ m. The non-zero, small minimum distance (0.005 m) is attributed to lattice interpolation artifacts inherent to the Immersed Boundary Method (IBM) used for the LBM coupling. When the solid boundary is mapped onto the discrete lattice nodes, a slight, instantaneous penetration can occur before the no-slip condition is enforced at the next time step. This is a common numerical effect, where the true continuous position of the robot momentarily falls slightly inside the collision boundary defined by the NMPC. This small residual distance is a numerical artifact equivalent to a $\approx 5.1\%$ error relative to the defined 0.10 m safety margin. Crucially, the presence of this small gap confirms that no sustained or problematic collision occurred, and the NMPC system successfully prevented the robot from achieving a zero or negative distance, i.e., complete overlap. Positive skewness values for all distance metrics show that closer proximity was less frequent, while flat kurtosis values suggest the distances were evenly distributed without extreme outliers.

The linear and angular velocity plots in Fig 3 below show the robot's speed and rotational adjustments during navigation. The linear velocity remains constant for most of the journey, suggesting steady forward motion. The angular velocity shows the changes when the robot adjusts its orientation, particularly during obstacle avoidance towards the end. During turbulent region traversal, the required corrective angular velocity reached the maximum allowable limit of the robot

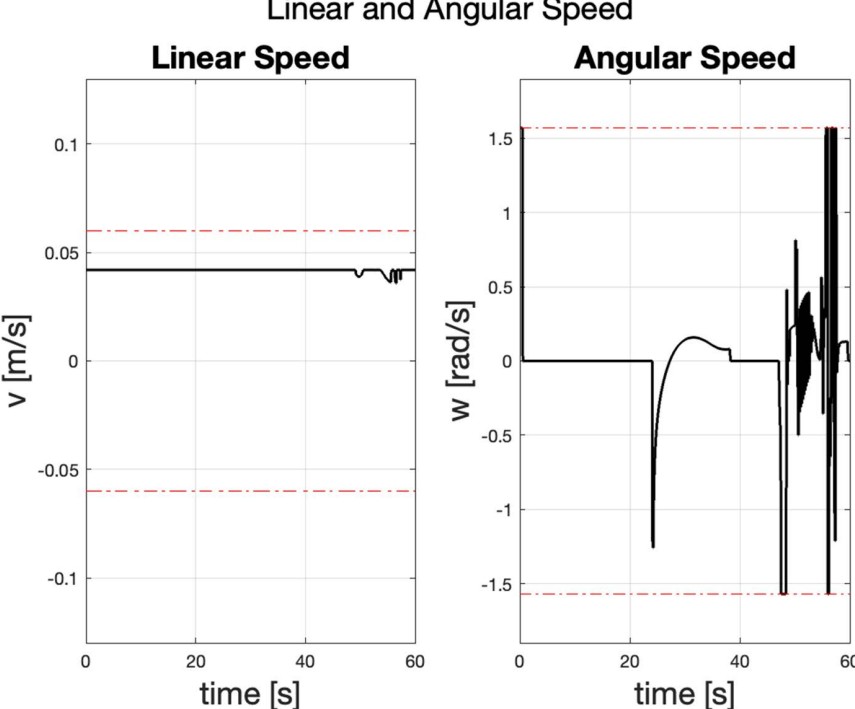

**Fig 3. Comparison of the robot's linear and angular velocities over time, illustrating the stability of linear motion versus the extreme fluctuations in angular velocity.**

model, suggesting that in a real-world scenario, the robot would have lost control of its orientation and path-following accuracy.

The Euclidean error shows the straight-line distance between the robot and the goal position over time. As expected in Fig 4, the error decreases steadily as the robot moves closer to the target. This trend indicates that the control algorithm effectively guides the robot toward the target. Minor errors in obstacle avoidance maneuvers.

The plot of the robot's trajectory in the *x-y* plane demonstrates the path followed by the robot during the simulation, as shown in Fig 5. The clear path shown in Fig 6 shows that the robot starts at the initial position (marked in green) and progresses to the goal position (marked in red). The trajectory reflects the robot's ability to navigate while avoiding the obstacles. The path appears smooth, indicating efficient trajectory planning with minimal deviations.

The minimum distance to obstacles in Fig 7 tracks the robot's proximity to the nearest obstacle during the simulation. The robot consistently maintains a safe distance from obstacles, with no values approaching zero, indicating successful collision avoidance. This behavior validates the obstacle avoidance strategy employed in the control algorithm. Table 8 shows the robot's performance in simulation, demonstrating high different path efficiency before 0.887. The actual path length was 2.828 m, which was longer than the optimal straight-line distance of 2.508 m, indicating that the robot performed well.

The shortest path distance, $d_{optimal} = 2.508$ m, is defined as the straight-line Euclidean distance between the robot's starting position ($P_{start}$) and the target position ($P_{target}$). This distance serves as a non-collision baseline ideal for evaluating the kinematic cost of the NMPC trajectory. It is important to note that this $d_{optimal}$ value does not account for the obstacle constraints and it is the absolute theoretical minimum distance in free space.

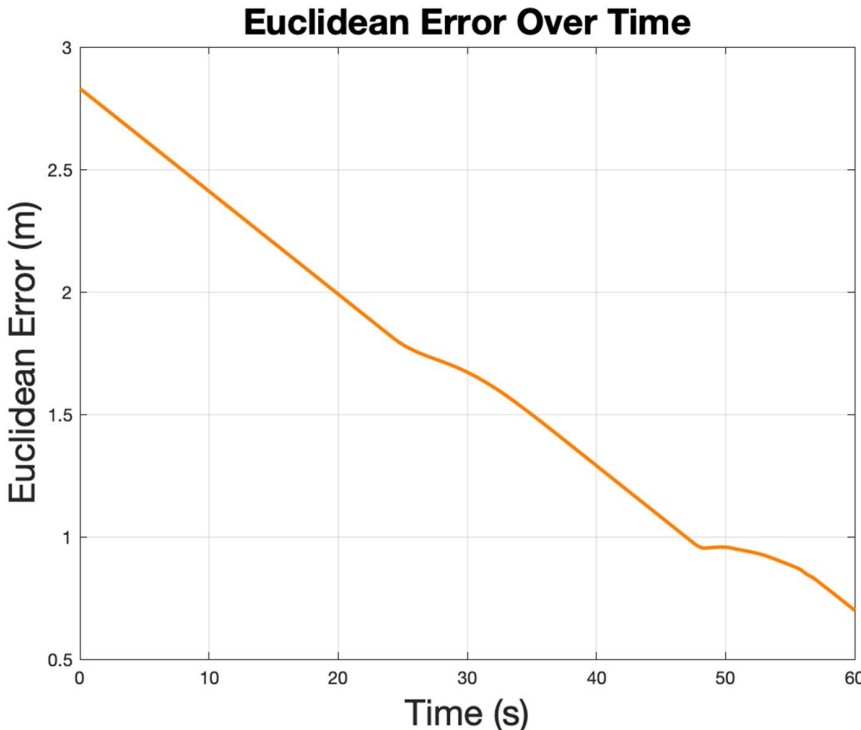

**Fig 4. Euclidean goal-directed error reduction illustrates the decreasing linear distance between the robot and the target location over time.**

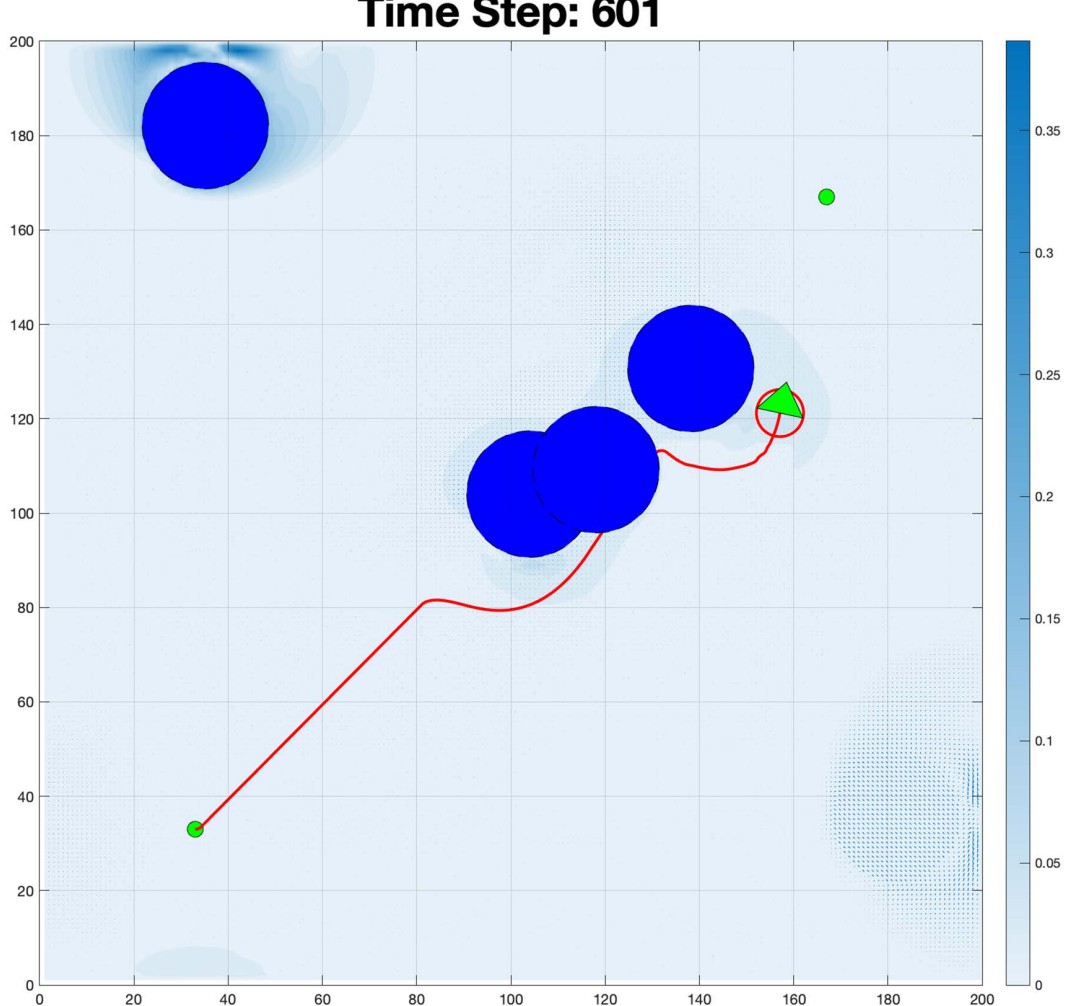

**Fig 5. Fluid-robot interaction in a dynamic environment: The color map represents fluid velocity magnitude (blue = high, white = low).**

The computational efficiency of NMPC algorithm is evident, with an average computation time of $5.91 \times 10^{-7}$ seconds per step and a total computation time of 0.0004 seconds, confirming its suitability for real-time trajectory planning. These results reflect a well-balanced approach to safe, efficient, and computationally feasible navigation.

The robot trajectory series plots in Fig 8 show effective navigation from the start to the goal while avoiding the obstacles. At $t = 0$, the robot is stationary, and by $t = 20$ seconds, it begins to move along a smooth path, maintaining safe distances from obstacles.

At $t = 25$, it adjusts its path to navigate through a cluster of obstacles, demonstrating precise maneuvering. By $t = 60$, the robot successfully achieves the objective without collisions, highlighting the robustness and efficiency of the control algorithm in dynamic environments.

## 7.2 CFD based spatial analysis

Table 9 presents key performance metrics for the fluid-robot interaction, highlighting how the robot navigates the fluid under different Reynolds number conditions. The Table 9 includes drag force components along the $X$ and $Y$ axes,

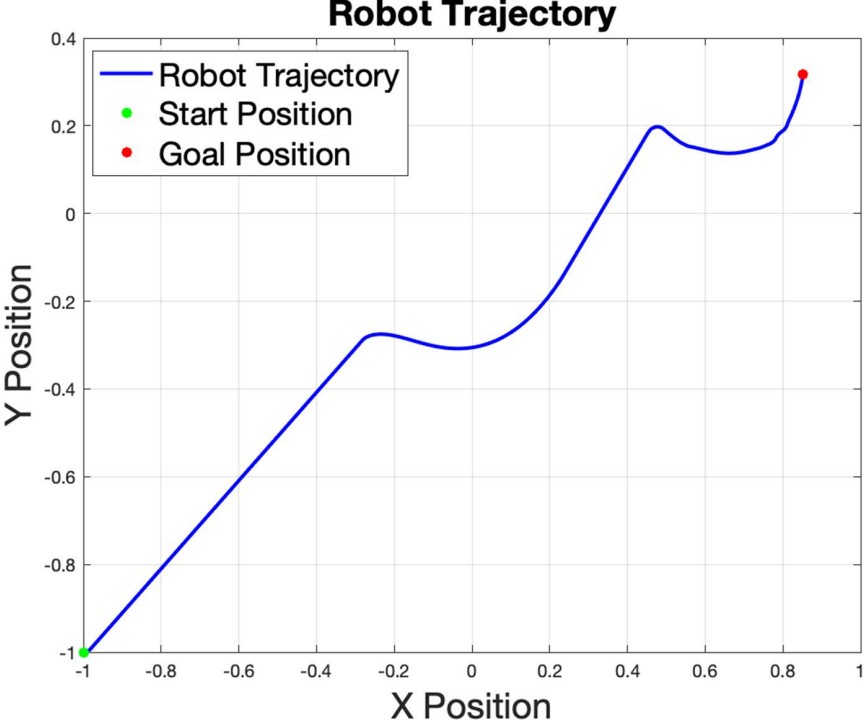

**Fig 6. The robot's trajectory from its initial point (green) to the target goal (red), demonstrating its ability to navigate while successfully avoiding the obstacles.**

representing the resistive forces acting on the robot. Energy dissipation quantifies the loss of energy due to fluid resistance, while the average velocity and the average vorticity describe the flow characteristics around the robot. Various Reynolds numbers indicate different flow regimes that affect the robot's movement and stability.

This data is further analyzed in subsequent plots to evaluate robot efficiency and the impact of fluid dynamics on its movement.

The Reynolds number plot in Fig 9 reveals significant variations between successive frames, with values starting low and rising as the simulation progresses. The initial low values suggest a laminar flow regime, as shown in Fig 10, while the spike and subsequent fluctuations indicate a transition to turbulent flow. In this scenario shown in Fig 11, the boundary condition was modified to allow fluid to flow from the left wall while also generating flow from an obstacle, creating a radial flow pattern.

The purpose of this adjustment was to observe how the system responds to different boundary constraints and fluid interactions. By setting the boundary condition to flow exclusively from the left wall, the fluid moves from left to the right, interacting with obstacles along its path. This creates turbulence, as shown in the simulation results. The visualization highlights areas of high velocity near the left boundary (in blue) and regions of recirculating flow around obstacles, demonstrating the impact of boundary conditions on fluid behavior. This simulation in time steps: 266 in Fig 12 shows the movement of the robot navigating through a turbulent fluid environment with obstacles. The color map represents the velocity of the fluid, with white indicating low-velocity regions and blue indicating high-velocity areas. At this stage, the robot has progressed significantly from its initial position (bottom left) towards its goal, following a trajectory (indicated by the red line). The fluid dynamics around the obstacles create complex vortices, especially near the clustered obstacles in the upper-mid region. The high-velocity zone around the robot shows that it is maneuvering through strong fluid forces, likely adjusting its

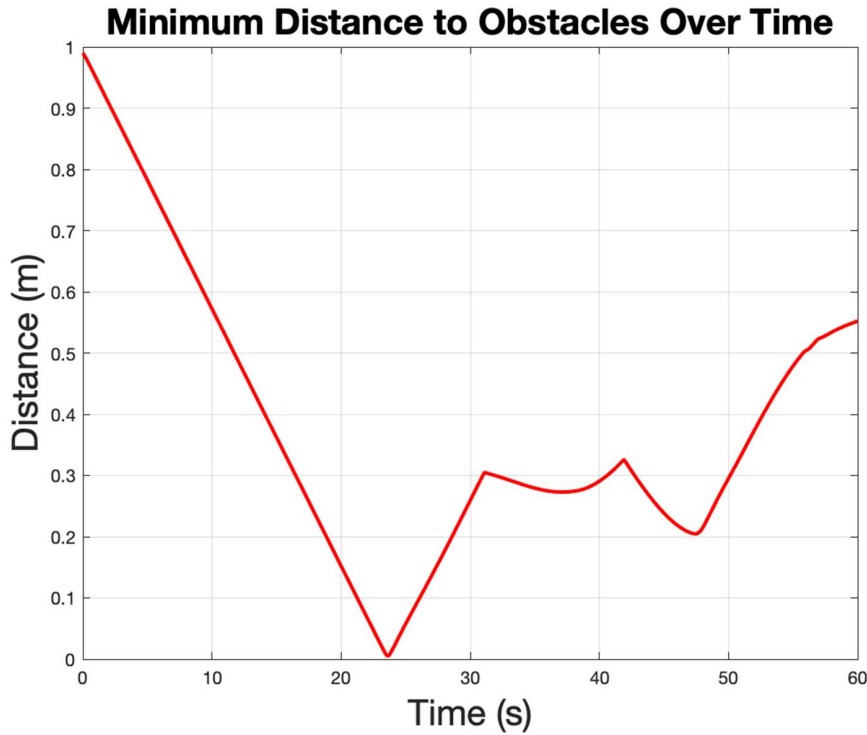

**Fig 7. Minimum distance to obstacle, which represents the obstacle avoidance safety margin.** It illustrates that the robot is maintaining distance from surrounding obstacles during the simulation.

**Table 8. Robot performance metrics of the robot in simulation.**

| Metric | Value |
|---|---|
| Optimal Path Length (m) | 2.508 |
| Actual Path Length (m) | 2.828 |
| Path Efficiency | 0.887 |
| Time to Goal (s) | 60.000 |
| Avg Computation Time (s) | $5.91 \times 10^{-7}$ |
| Total Computation Time (s) | $4.00 \times 10^{-4}$ |

path to avoid collisions with the obstacles. This evolution highlights the robot's influence on the surrounding fluid and how it modifies the flow conditions as it moves. Such transitions may impact the robot's performance and stability in dynamic environments.

The drag forces in the $X$ and $Y$ directions start at zero and increase sharply during the initial frames as the robot begins to interact with the fluid. The plot in Fig 13 shows that the drag force $X$ is significantly larger than the drag force $Y$, indicating that most of the resistance experienced by the robot is aligned with its primary motion direction. Following the initial steep rise, the drag forces stabilize, reflecting the robot's steady velocity and maintaining efficient movement. The initial spike in $Y$ is a result of transient hydrodynamic forces when the robot begins its first forward and turning, not a continuous moving in the $Y$ direction. The fact that the line immediately flattens proves the robot does not continue moving in the $Y$ direction.

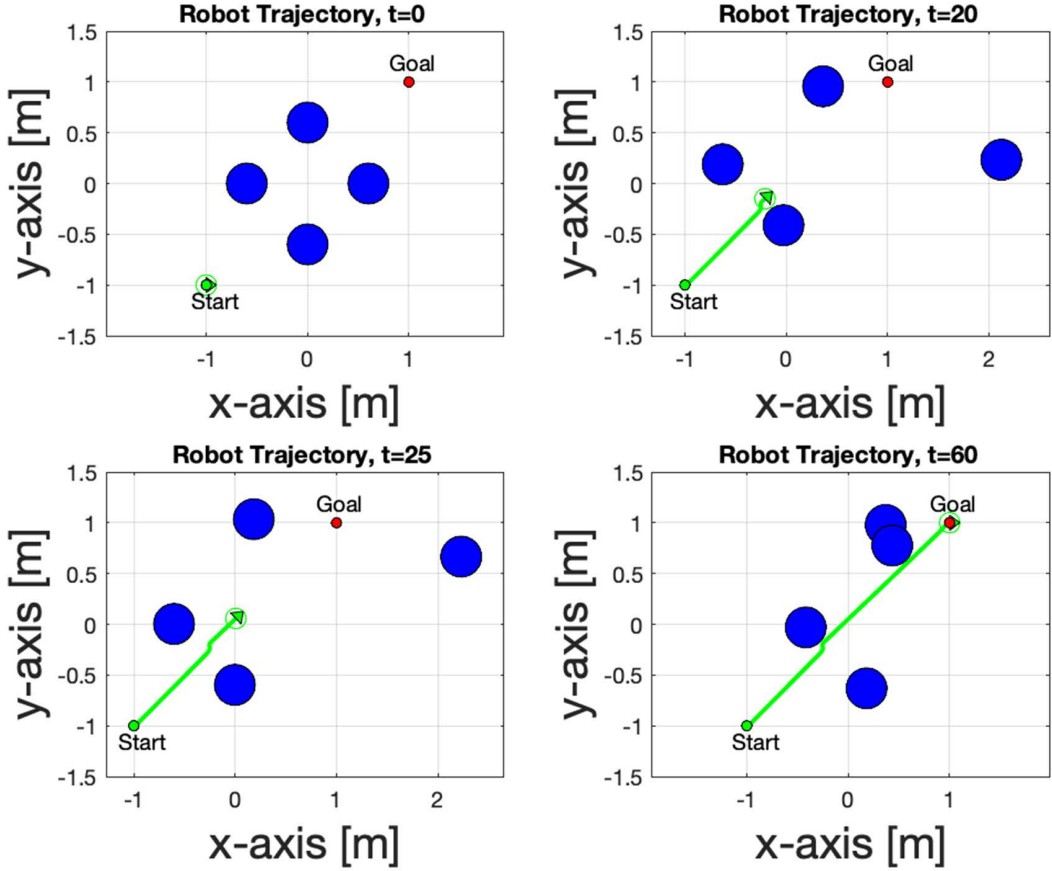

**Fig 8. The temporal evolution of the navigation trajectory, which shows the robot's path at various time steps.**

**Table 9. Fluid dynamics parameters calculated from LBM simulations.**

| Reynolds Number (Re) | Drag Force $X$ (N) | Drag Force $Y$ (N) | Energy Dissipation (W) | Avg Velocity (m/s) | Avg Vorticity ($s^{-1}$) |
|---|---|---|---|---|---|
| 0.221 | 0.466 | 0.0026 | 0.301 | 0.0004 | 0.0002 |
| 0.470 | 0.450 | 0.0164 | 0.474 | 0.0009 | 0.0001 |
| 0.714 | 0.435 | 0.0522 | 0.949 | 0.0014 | 0.0002 |
| 0.221 | 0.388 | 0.0946 | 0.301 | 0.0004 | 0.0002 |
| 0.470 | 0.372 | 0.112 | 0.474 | 0.0009 | 0.0001 |
| 0.714 | 0.357 | 0.129 | 0.949 | 0.0014 | 0.0002 |
| 0.961 | 0.341 | 0.146 | 1.17 | 0.0019 | 0.0002 |

The energy dissipation seen in Fig 14 shows the rate at which kinetic energy is lost in the fluid due to viscous effects. The initial values are minimal, corresponding to the laminar flow regime. As the Reynolds number increases and turbulence develops, energy dissipation spikes occur dramatically, particularly at specific frames where instabilities are observed. The peak energy dissipation indicates regions of high turbulence or flow instability, which may necessitate adjustments to the robot's design or control strategies to minimize energy losses and enhance efficiency.

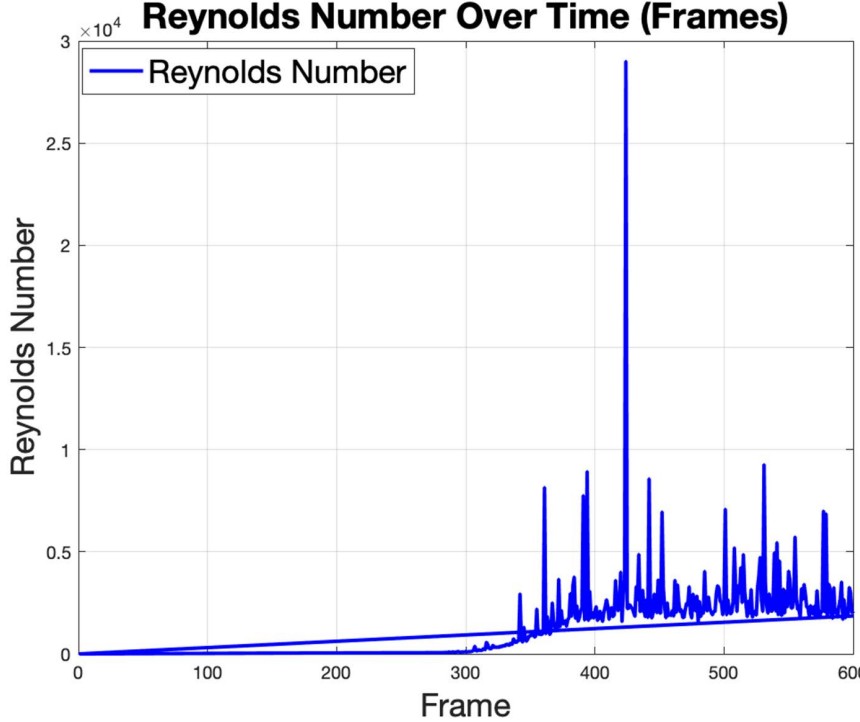

**Fig 9. The Reynolds number evolution during robot navigation, which shows significant variations across simulation frames, with values starting low and increasing over time.**

The average velocity seen in Fig 15 reflects the overall speed of the fluid, which increases steadily as the simulation progresses, likely due to the movement of the robot and its impact on the fluid flow.

The spikes in velocity suggest regions of intensified flow, possibly resulting from wake-up effects or interactions with obstacles. The average vorticity plot complements this by showing the rotational dynamics of the fluid. The higher vorticity values correspond to the formation of vortices behind the robot, indicating wake turbulence, as shown in Fig 16.

The qualitative features of the vorticity contours in Fig 12, serve as a crucial proxy for the unquantified pressure distribution and drag forces acting on the robot. Specifically, the wake, the region immediately downstream of the body, is characterized by strong shear layers and the rapid diffusion of vortex structures. The size and intensity of this wake are directly proportional to the magnitude of the low-pressure region forming behind the body, which dictates the pressure drag. The NMPC actively controls the trajectory to minimize the size of this low-pressure, high-vorticity zone. The resulting small, diffuse vortices indicate that the controlled motion is effectively minimizing the time-averaged pressure difference between the front and rear faces of the robot. This successful active shaping of the wake is essential for limiting the net resistance (drag) and thereby validates the NMPC's role in optimizing the robot's path-following energy expenditure in highly dynamic fluidic environments.

### 7.3 Comparison with existing studies

This work significantly contributes to mobile robot navigation by addressing a critical gap in recent research. While existing studies have focused on disturbance rejection or effective path planning, they often overlook the impact of fluid dynamics on robot performance. To highlight the value of our analytical framework, we provide a comparison in Table 10,

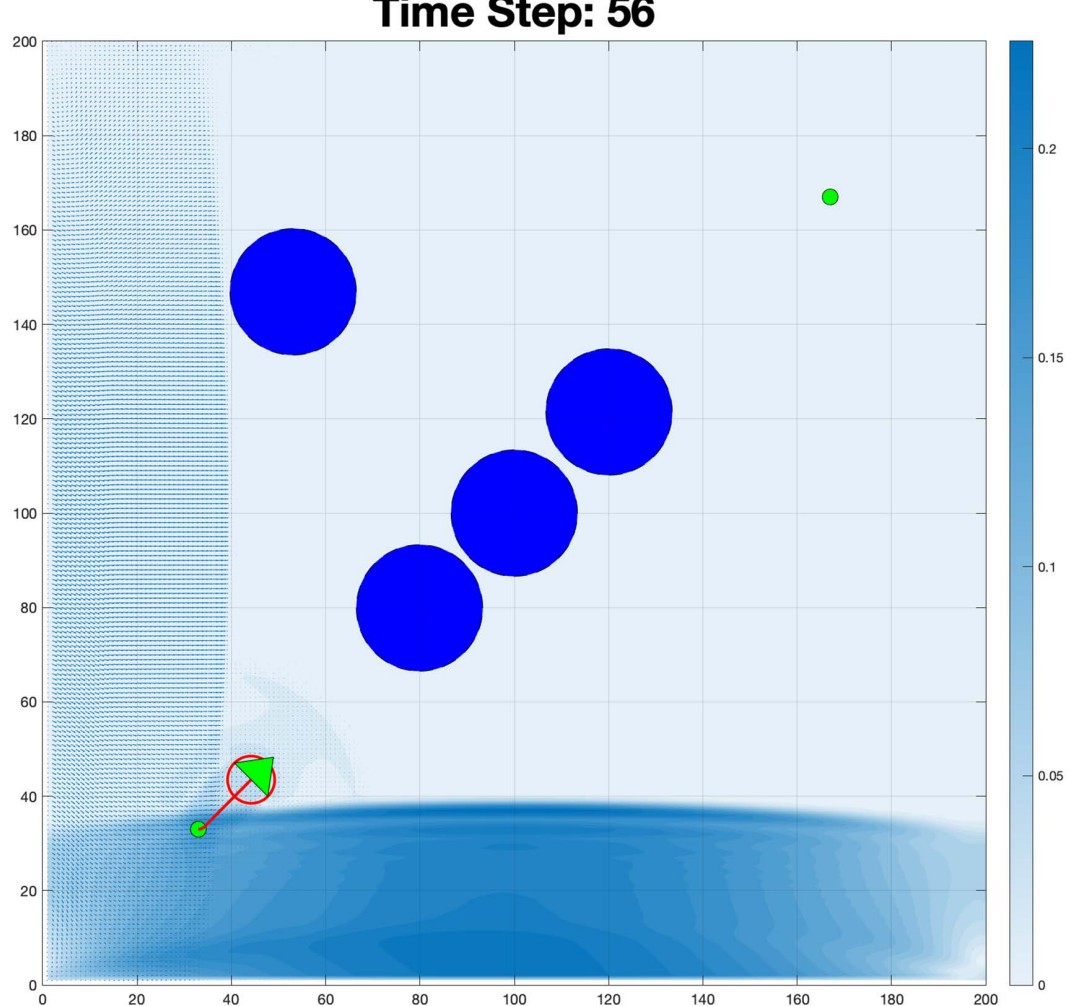

**Fig 10. An initial laminar flow regime (low values) resulting from the fluid interaction pattern at the bottom.**

which delineates the techniques, key objectives, and significant outcomes of each study, enabling a clear contrast with our current research.

Our study is distinguished by its innovative analytical framework that integrates a navigation technique with an in-depth fluid dynamics analysis, offering novel insights into the interaction between fluid environments and robotic mobility. The NMPC-planned trajectory achieves a high path efficiency of 0.887 and a short actual path length of 2.828 m, demonstrating its navigational efficiency. Its low computation time confirms NMPC's suitability for real-time trajectory planning.

Our framework's distinctive value lies in its capacity to achieve robust performance while also offering essential post-hoc analysis of fluid forces, a feature lacking in most similar studies. This allows us to measure the actual energetic expense and stability of the trajectory, representing a substantial improvement over purely kinematic or simplified dynamic models.

The LBM analysis of the Reynolds number progression from laminar to turbulent regimes highlights the robot's influence on flow dynamics and the necessity for robust stability measures in high-turbulence conditions. Drag forces

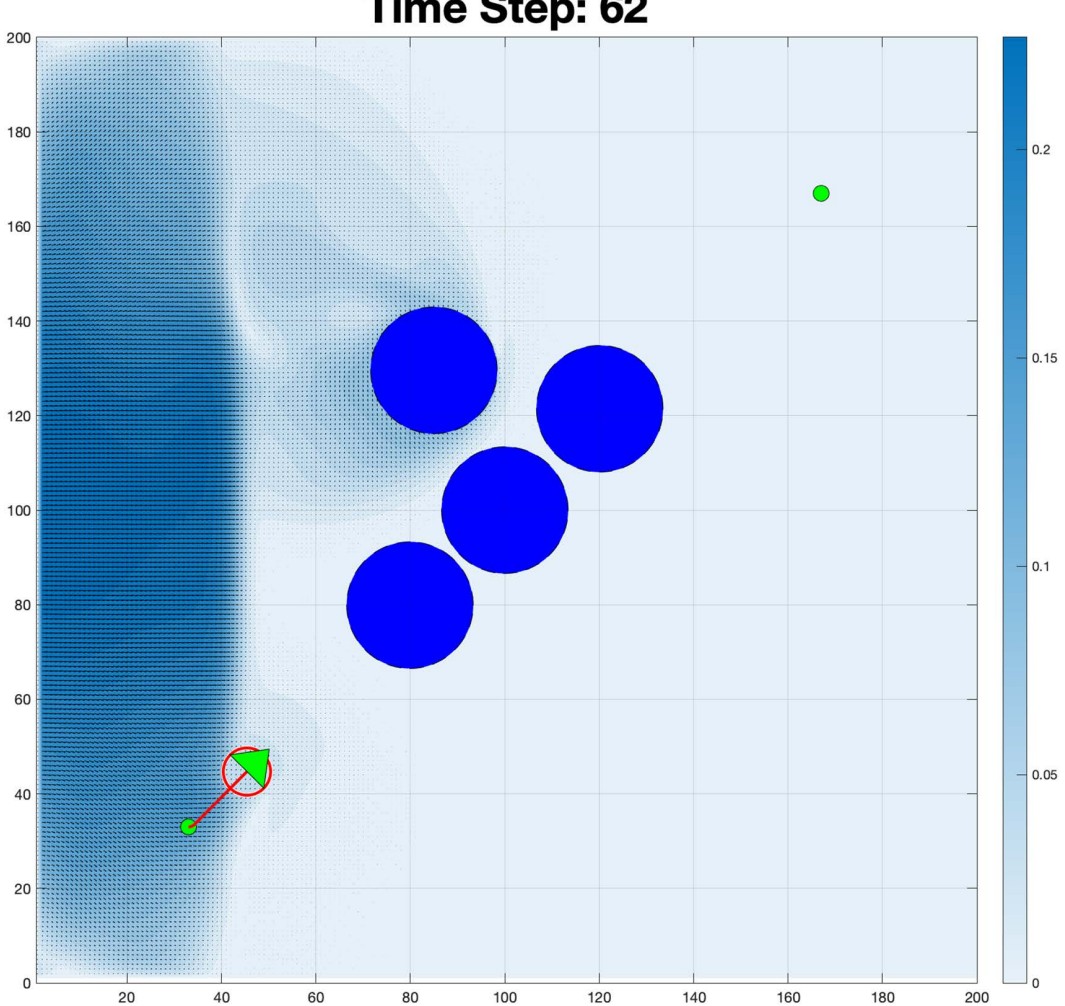

**Fig 11. Boundary-driven fluid flow patterns: Color gradients represent velocity magnitude (blue = high, white = low).**

underscore the significant resistance faced by the robot, while their eventual stabilization demonstrates the robot's controlled motion and its ability to maintain efficient movement. Energy dissipation trends reveal critical regions of turbulence, indicating areas where energy losses can be minimized through optimized design [21]. Additionally, the average velocity and vorticity plots illustrate intensified flow and rotational dynamics, pointing to wake turbulence and vortex formation. These findings collectively underline the significance of integrating navigation and fluid interaction analyses for designing efficient and robust robotic navigation systems [22].

## 8 Advantages of LBM method

This study employs the LBM for modeling fluid dynamics. Both LBM and classic techniques such as the finite volume method (FVM) are CFD approaches employed to resolve the Navier-Stokes equations, however they fundamentally differ in their methodologies. Conventional CFD techniques describe fluid dynamics in continuum mechanics by solving partial differential equations (PDEs), which can be computationally intensive and challenging to resolve for intricate geometries and turbulent flow. These PDEs control multiple variables, such as velocity and pressure [54].

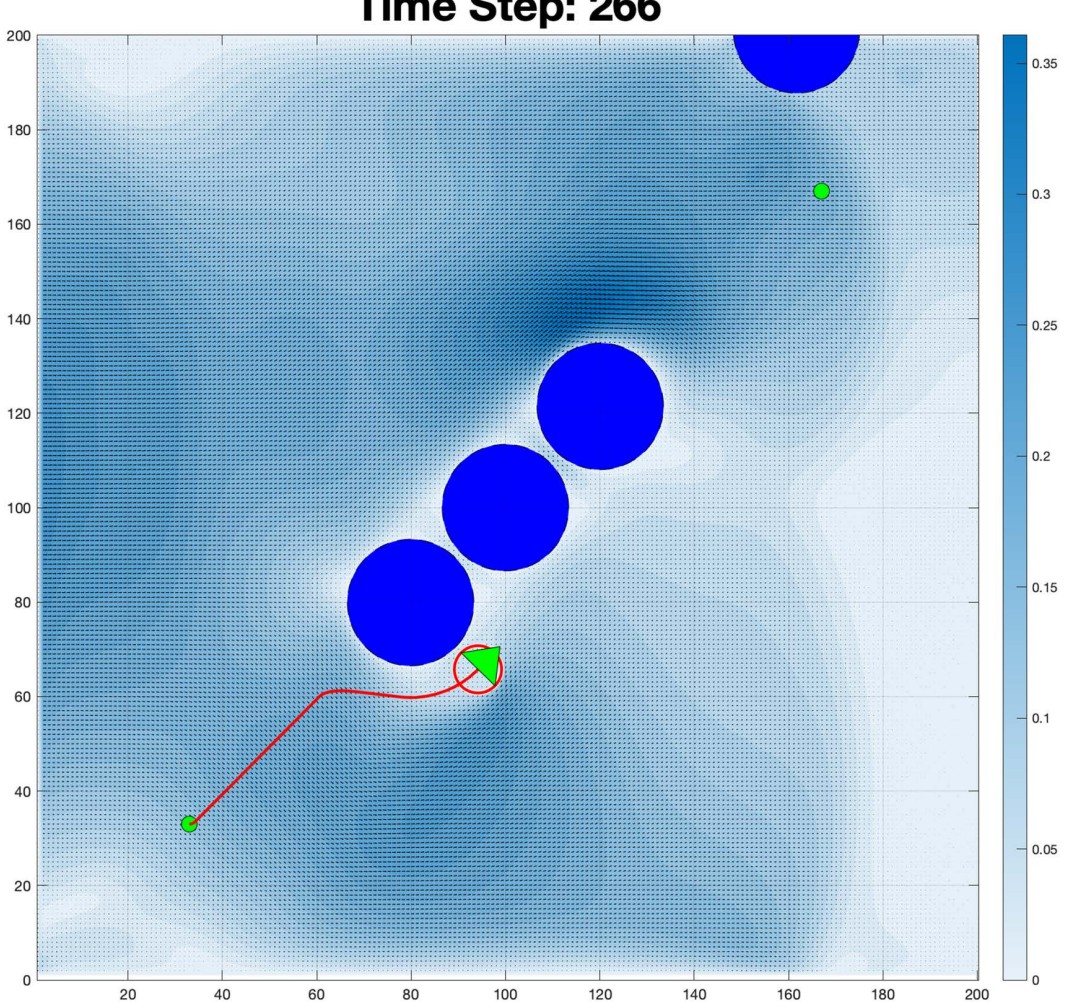

**Fig 12. Complex fluid-robot interaction at time step 266: Color-coded velocity magnitude (white = low to blue = high) with directional vectors illustrating complex fluid dynamics, including vortices and recirculation zones around the obstacles.**

They are based on principles of Newtonian physics and involve solving nonlinear coupled PDEs, which can be quite difficult, especially in situations that involve intricate geometries and turbulent flow. Fluid simulations employ several CFD techniques, such as the finite volume method (FVM), finite difference method (FDM), and finite element method (FEM). The research utilized LBM due to its high level of computing efficiency, scalability, and effectiveness in dealing with complex geometries. A significant advantage of LBM is its capacity to handle the complex interactions between fluids and solid materials effectively. Unlike FVM and FDM, which approach the Navier-Stokes equations from a larger perspective, LBM operates at a more granular, mesoscopic level by using particle distribution functions. This capability enables LBM to accurately replicate small-scale flow behaviors, like wake turbulence and vortices, making it particularly well-suited for robotic navigation in fluid settings [55]. Moreover, LBM performs exceptionally well with parallel processing, which boosts its speed and efficiency when compared to FEM, known for demanding a solution of large linear systems. Its lattice design further enhances its effectiveness in high-performance computing settings, significantly reducing the time needed for extensive simulations.

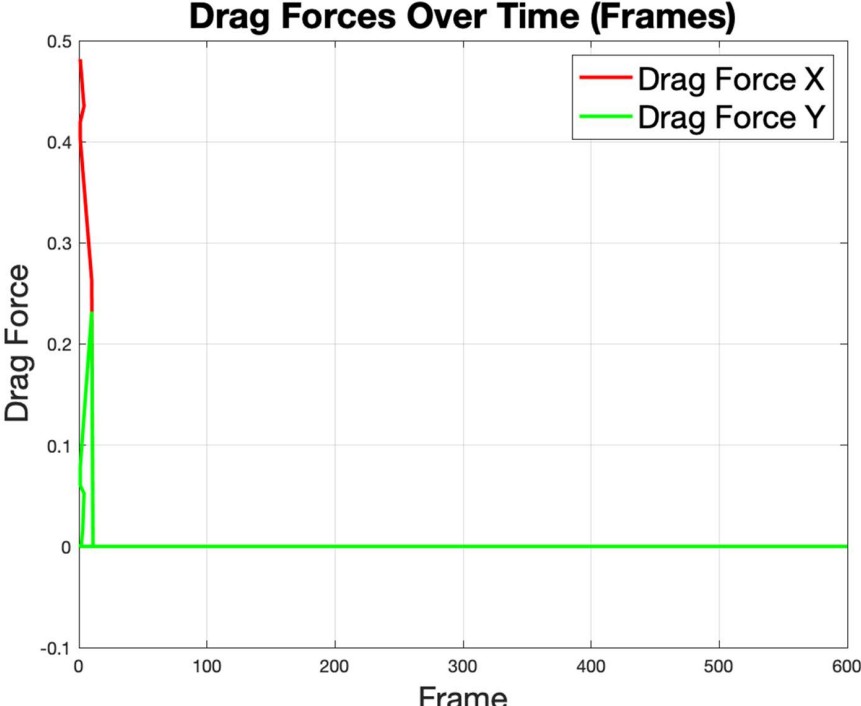

**Fig 13. Directional drag forces on the robot, showing an initial increase in the *X* and *Y* components due to robot-fluid interaction at the start of the simulation.**

## 9 Advantages of NMPC over traditional control methods

NMPC has several benefits over other traditional control methods in dynamic and complex settings, making it a good choice for mobile robot navigation in this study. Unlike classical controllers like PID or active techniques, NMPC takes into account system constraints, nonlinear dynamics, and predictions of future states, which helps in identifying more optimized and collision-free paths. Furthermore, in contrast to approaches reliant on RL, NMPC provides deterministic solutions that can be analytically validated without requiring significant training datasets. Another notable benefit of NMPC is its effectiveness in managing external disturbances and environmental constraints. Fluid interactions can generate vortices, turbulence, and drag forces that unpredictably influence the robot's movement. Although PID controllers occasionally encounter difficulties in adapting to fluctuations induced by fluid dynamics, the predictive characteristics of NMPC enable it to devise smooth, robust paths that may automatically mitigate these effects [56].

## 10 Autonomous navigation in Fluid Environments vs. Land Environments

Autonomous robots functioning in fluid environments encounter significantly different challenges than those traversing terrestrial terrains [14]. Terrestrial robots interact with fixed or organized terrains, where locomotion mostly depends on friction and traction. In contrast, fluid-based robots traverse a dynamic and constantly evolving medium, where forces such as drag, buoyancy, and wake turbulence influence their motion. External factors produce unexpected disturbances, requiring advanced control systems to maintain stability and efficiency. Land robots utilize wheels or legs for locomotion, whereas fluid robots encounter persistent opposition from the surrounding fluid. This persistent opposition necessitates advanced techniques for forecasting their trajectories. Moreover, maneuvering around obstacles in fluid environments is more intricate due to the necessity of accounting for wake turbulence and disruptions induced by the flow. In contrast to

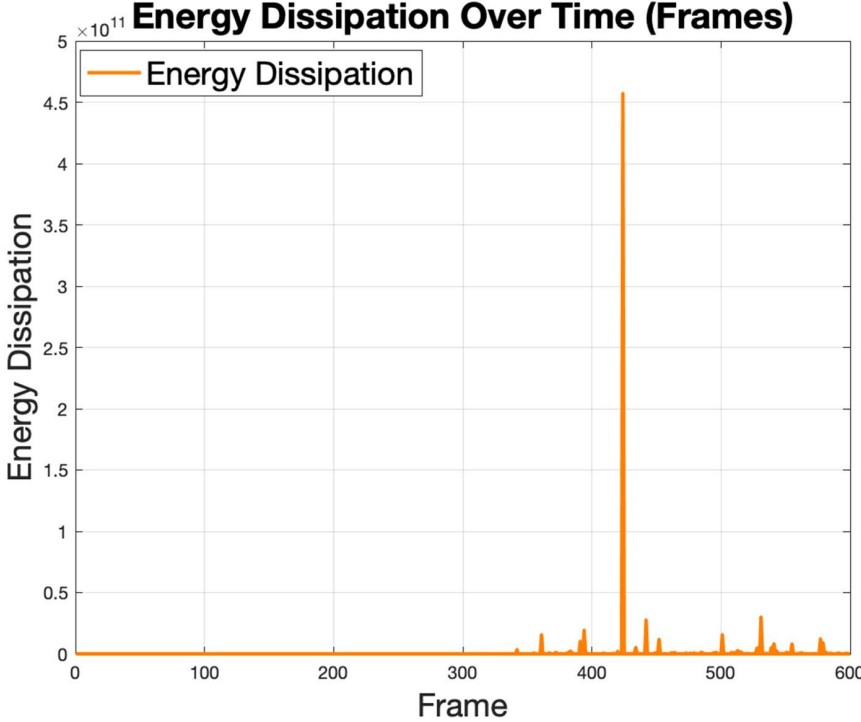

**Fig 14. The fluid energy dissipation profile shows a transition from low dissipation during laminar flow to high dissipation during turbulent flow.**

land robots that depend on direct sensor feedback, robots operating in a fluid medium must also consider fluid-induced drifts that dynamically influence their position over time. The predictive features of NMPC approach make it an ideal option for addressing these challenges, since it facilitates effective navigation for robots while accounting for dynamic barriers. A thorough comprehension of the intricate fluid interactions influencing the robot's motion necessitates a more in-depth analysis than what NMPC model alone offers.

## 11  Impact of fluid interactions on robot motion

The hybrid simulation framework reveals deeper insights into the impact of fluid–structure interaction on the robot's trajectory, control effort, and stability. While the robot demonstrates successful navigation and convergence toward the goal, the recorded data indicates that fluid dynamics, especially under high Reynolds number conditions, where the flow transitions from laminar to turbulent, introduce substantial challenges to robot stability and control. In such regimes, the fluid behavior becomes increasingly unsteady, leading to phenomena such as vortex shedding, fluctuating pressure zones, and asymmetric drag forces. These effects introduce complex, time-varying disturbances to the robot, which can destabilize its motion, increase energy consumption, and demand higher control effort from NMPC to maintain trajectory accuracy and obstacle avoidance performance [18]. For instance, at timestep 266 frames, as illustrated in Fig 12, the robot enters a highly dynamic fluid region marked by dense vortex structures and elevated vorticity. This portion of the domain lies downstream of several closely spaced obstacles, where the fluid flow becomes disrupted and transitions into a state of unsteady turbulence. The formation of alternating vortices in the robot's wake creates localized zones of low and high pressure along the body's lateral surfaces. These unbalanced pressure distributions exert oscillatory lateral forces on the robot, inducing minor angular deviations and fluctuations in yaw. Such a phenomenon is indicative of a classical

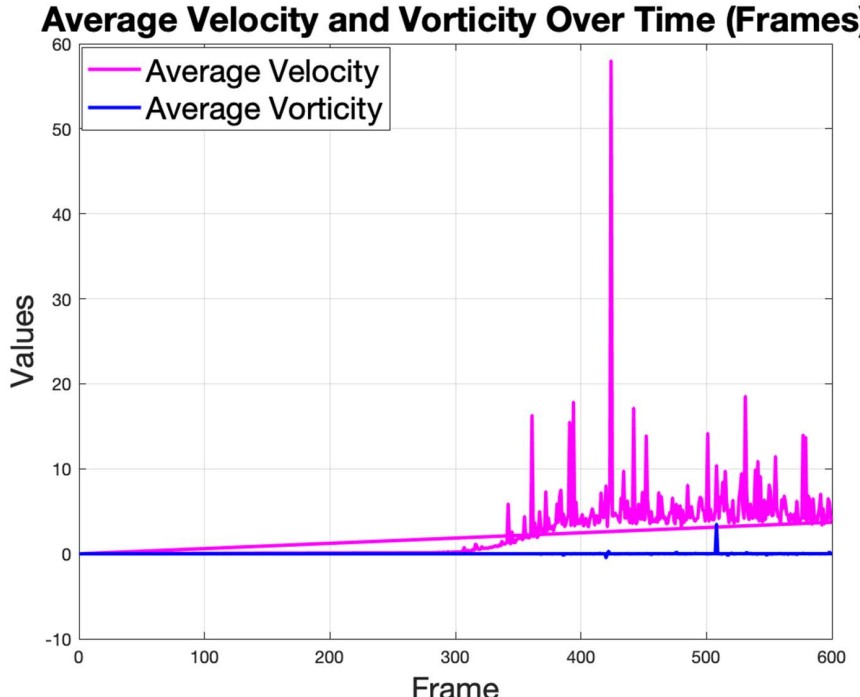

**Fig 15. The average fluid velocity increases over the simulation, influenced by the movement, and the average vorticity illustrates the fluid's rotational dynamics.** The most striking observation is the time history of the average velocity, which does not follow a single path but separates into two distinct, stable ranges, a phenomenon known as bimodal oscillation. This indicates the fluid rapidly cycles between two energetic states, likely transitioning between its high-power thrust generation phase and a lower-power recovery stroke.

vortex-induced vibration (VIV) response, commonly encountered in fluid–structure interaction (FSI) systems where bodies move through or remain stationary in turbulent flow fields [57]. In the context of this simulation, the robot responds to these forces by initiating frequent and sharp corrective rotations observable as a spike in angular velocity in Fig 3 and supported by the peak values listed in Table 6. These corrective actions are a direct result of NMPC attempting to stabilize the robot's pose amidst external disturbances caused by unsteady flow separation. The presence of these instabilities not only impacts motion stability but also increases the control system's computational load, as NMPC must resolve more aggressive trajectory corrections within each prediction horizon. In practical scenarios, if unaccounted for, such conditions could lead to cumulative yaw drift, loss of localization, or inefficient energy usage. The observation at the timestep 266 frames therefore serves as a strong example of how localized fluid dynamics phenomena such as wake turbulence and vortex shedding can temporarily compromise trajectory fidelity and demand significant control input to restore motion.

Furthermore, under radial inflow boundary conditions in Fig 11, the robot exhibits a noticeably higher standard deviation in angular velocity, indicating less stable rotational motion. The robot's unstable behavior under radial inflow conditions can be traced back to how the fluid enters the domain. Instead of coming from one direction, like in the left-wall inflow case, the fluid in the radial setup approaches from all sides of the obstacle region. This means the robot is surrounded by flow that is constantly changing direction and intensity. As it moves, it is hit by drag forces from different angles, and this throws off its orientation slightly, forcing the controller to keep correcting. These corrections show up as fluctuations in angular velocity, as seen in Fig 3. On the other hand, with unidirectional inflow from the left wall, the fluid moves more uniformly. The robot can move through it with fewer surprises. The velocity field is consistent, the drag is more predictable, and the controller does not need to exert as much effort to maintain the robot's stability. This contrast makes it clear

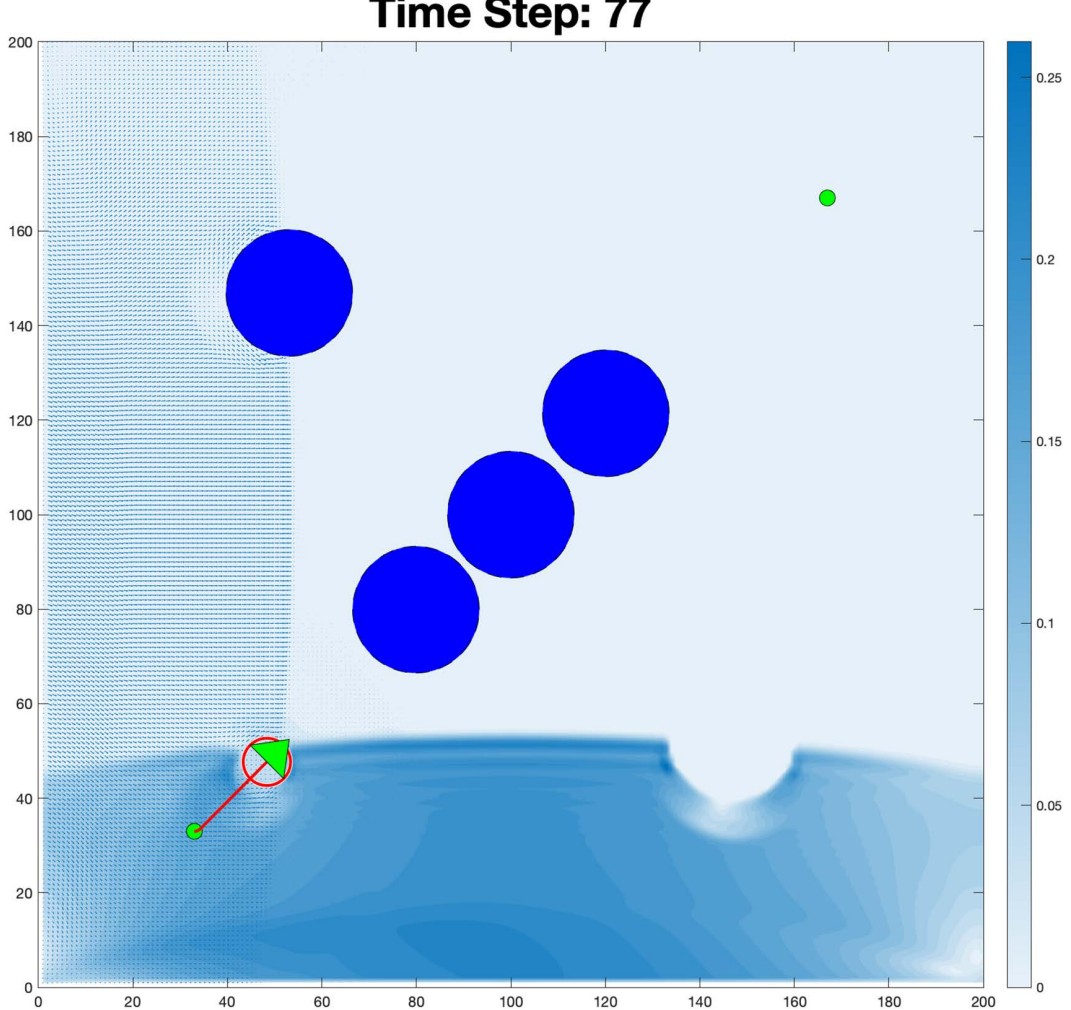

**Fig 16. Spatial visualization illustrating the formation of wake turbulence vortex behind the robot, resulting from the robot's movement.**

that the direction of fluid inflow has a major impact on how much effort the robot needs to stay on track. There were also times during the simulation when the robot hit its maximum allowed angular velocity of ±1.57 rad/s. These spikes mostly happened in areas with strong vortices or right near obstacles, where the fluid flow became highly unsteady. When this happens, it means the controller was basically maxed out, and it could not do much more to help the robot stay oriented. That is not ideal, especially in turbulent or unpredictable environments where the robot might still need to respond to new forces. It highlights a limitation in using fixed control parameters when the environment itself is constantly shifting. Looking ahead, the simulation could be expanded by digging deeper into the generated data. For instance, it would be useful to check how the robot's turning behavior (curvature) relates to local vorticity in the fluid. That could tell us exactly how turbulent zones affect the robot's ability to steer. It would also help to divide the domain into zones based on how intense the flow is and then see where the controller is working the hardest. This study aims to enhance the precision of performance assessments for autonomous robotic systems in fluid environments Utilizing an innovative two-stage analytical framework, we present a methodology for evaluating the resilience of control systems to intricate fluid-structure interactions. This methodology is crucial for the dependable validation of control algorithms prior to the actual implementation of

**Table 10. Methodological comparison of robotic navigation studies.**

| Study | Methodology | Objective | Key Findings |
|---|---|---|---|
| DeSouza & Kak (2002) [49] | Vision-based navigation in structured and unstructured environments. | Survey of vision systems for mobile robot navigation. | Identified challenges and solutions in implementing vision systems for both indoor and outdoor navigation, highlighting the importance of environment structuring. |
| Zhangfan LU & Ran Huang (2021) [50] | The study employs deep reinforcement learning (DRL) for navigation. | Aiming to navigate unknown environments without prior maps. | Navigating safely in unstructured environments, the developed network, LDDPG, not only outperforms ADDPG and DDPG in reward convergence but also selects shorter paths compared to ADDPG, all achieved through unsupervised training in dynamic environments. |
| Maram Ali & Saptarshi Das (2023) [51] | Area exploration using random-walk. | Determine the effectiveness and cost-efficiency of using a random-walk search algorithm for a multitude of mobile robots to investigate an area. | Found that increasing the number of robots in a swarm significantly reduced the area exploration time; furthermore, the cost-performance analysis of the swarm was effective in determining the optimal robot number. |
| Wan Xu & Dongting Liu & et al. [52] | Introduces an improved optimization algorithm, LAPGWO, for path planning of multiple mobile robots. | Improve the efficiency and effectiveness of path planning for multiple mobile robots. | Demonstrating improved path planning efficiency with reductions in shortest path and calculation time when compared to the traditional GWO algorithm, the LAPGWO algorithm also exhibited enhanced convergence and stability. |
| Xingshuo Hai & Ziming Zhu & et al. (2025) [53] | Employing the MOSFMO algorithm for multi-objective path optimization (considering path length and safety) and managing coordination through multiple candidate paths. | Develop a dual-layer decision-making framework for the trajectory of autonomous mobile robots. | The proposed decision-making framework achieves up to 99% on-time arrival rates in dynamic environments, demonstrates an 11% increase in mission success compared to existing methods, and adapts to unforeseen events in complex scenarios. |
| This Study | Uses a decoupled, two-stage process where the NMPC generates the trajectory first, and the LBM analyzes it later. | Assessing the impact of fluid dynamics on navigation performance. | NMPC, utilizing a kinematic model, exhibited effective path planning and obstacle avoidance, attaining a path efficiency of 0.8869. The next fluid dynamics analysis, however, offers a crucial assessment of this kinematically optimal trajectory, exposing the influence of drag forces and energy dissipation that were overlooked during the planning phase. |

autonomous robots in genuine underwater settings, where dynamic and unpredictable fluid dynamics present an ongoing challenge [58].

## 12 Conclusions and future work

The analysis of the robot's navigation in a fluid environment was conducted through a two-phase simulation, supported by a rigorous grid independence study to ensure the physical integrity of the results. This demonstrates its effectiveness in achieving smooth trajectory planning, obstacle avoidance, and convergence. Statistical evaluation of key parameters, including Euclidean error, velocity, and distances to obstacles, validates the robustness of NMPC control algorithm. Validation against a $200 \times 200$ benchmark grid confirms that the control inputs are derived from a converged hydrodynamic field, where numerical diffusion is minimized and the viscous boundary layer is resolved by 3–5 lattice units. Consequently, the robot maintained stable motion and consistently avoided collisions, reflecting precise navigation even in complex and dynamic environments. LBM analysis presented information on the fluid dynamics around the robot, highlighting the impact of Reynolds number, drag forces, energy dissipation, and vorticity. The use of a high-resolution lattice was critical for capturing the high-frequency vortex shedding and shear layer gradients that characterize these regimes. The progression from laminar to turbulent flow and the identification of regions with high drag and energy loss underscore the importance of optimizing the robot's design and motion to enhance efficiency, especially where the application of the robot is either local or global. Notably, metrics such as, a high path efficiency of 0.887 and a peak memory footprint of only 2.88 MB confirm the viability of the algorithm for real-time deployment. This minimal computational overhead suggests that the

framework is uniquely suited for integration into embedded systems and low-power onboard microcontrollers in autonomous underwater or aerial vehicles, where the hardware resources are strictly limited.

Future works may include:

- Optimizing the robot geometry to minimize drag forces and reduce energy dissipation, thereby improving overall performance in fluid environments.

- Strategies for managing turbulence and wake dynamics, such as implementing flow control techniques or adjusting motion parameters, should be explored to improve stability and efficiency.

- Experimental studies in physical environments, such as underwater or aerial scenarios, should be conducted to validate the findings of the simulations.

- Navigation in multiphase reacting fluid environments.

- Continuation of this work in future will include the external validation of the LBM solver. This will involve simulating the canonical case of flow past a circular cylinder **Re=100** to precisely calculate and benchmark the drag coefficient ($C_D$), lift coefficient ($C_L$), and Strouhal number (**St**) against literature values, ensuring an accuracy within the required $\leq 5-10\%$ error margin.

- The framework necessitates full statistical validation of the NMPC performance to demonstrate robustness. This requires running the planner across multiple randomized trials to calculate the mean ($\mu$) and standard deviation ($\sigma$) for all key metrics.

- The current work establishes the analytical power of the hybrid NMPC-LBM framework, while a comprehensive quantitative baseline comparison is a critical next step. This analysis will involve running the NMPC against PID, LQR, and RL controllers under identical fluid conditions and obstacle configurations.

- Extending the simulation cases to include $Re > 10^4$ will require integrating a turbulence model, e.g., large eddy simulation or a specific $\tau$ model into the LBM. This is essential for clarifying the framework's applicability in highly turbulent industrial or oceanic scenarios. We will incorporate viscosity models (such as the Carreau or power-law model) into the LBM. This will clarify the framework's scope for analyzing robot performance and energy metrics in non-Newtonian environments, such as industrial slurries or deep-sea muds.

## 13 Acknowledging the limits

Despite the successful demonstration of the NMPC-LBM framework, it is crucial to recognize the inherent simplifications and limitations of the decoupled methodology before drawing generalized conclusions:

- The results demonstrate that while NMPC provides a near-optimal kinematic path (efficiency 0.887), the decoupled fluid analysis reveals a significant challenge. The required control effort for dynamic stability is non-trivial and, at times, saturates the control input.

- This finding invalidates the direct translation of the NMPC trajectory to a real-world fluid environment without the inclusion of a dynamic model in the control loop.

- The current study models the robot as a simplified shape, e.g., a circle or cylinder, and does not account for complex, high-frequency body oscillations or deformable surfaces.

- We explicitly acknowledge that the framework utilizes a two-stage, decoupled analytical design where the NMPC trajectory is kinematically determined a priori, and the hydrodynamic forces are calculated a posteriori using the LBM. This

design is highly effective for its intended purpose for assessing the performance envelope and energy cost of a kinematically optimal plan.

## Supporting information

**S1 Data We have provided the minimal data set as Supporting information in a single zipped folder labeled S1 Data. This archive contains two Excel files: `Fluid_Dynamics_Analysis.xlsx', which includes the raw hydrodynamic results (Reynolds numbers, drag coefficients, and vorticity values), and `Simulation_Metrics.xlsx', which provides the control performance data (path efficiency, Euclidean error, and computation times) used to generate the figures and statistics in the paper.**
(ZIP)

## Acknowledgments

For the purpose of open access, the author has applied for a Creative Commons Attribution (CC BY) license to any Author Accepted Manuscript version arising from this submission.

## Author contributions

**Conceptualization:** Maram Ali, Saptarshi Das, Stuart Townley.

**Data curation:** Maram Ali.

**Formal analysis:** Maram Ali.

**Funding acquisition:** Maram Ali, Saptarshi Das.

**Investigation:** Maram Ali.

**Methodology:** Maram Ali, Saptarshi Das, Stuart Townley.

**Project administration:** Saptarshi Das, Stuart Townley.

**Resources:** Saptarshi Das.

**Software:** Maram Ali.

**Supervision:** Saptarshi Das, Stuart Townley.

**Validation:** Saptarshi Das, Stuart Townley.

**Visualization:** Maram Ali.

**Writing – original draft:** Maram Ali.

**Writing – review & editing:** Saptarshi Das, Stuart Townley.

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
