## [Decision Letter · Decision Letter 0]

10 Nov 2025

Dear Dr. Das,

Thank you for submitting your manuscript to PLOS ONE. After careful consideration, we feel that it has merit but does not fully meet PLOS ONE’s publication criteria as it currently stands. Therefore, we invite you to submit a revised version of the manuscript that addresses the points raised during the review process.

If applicable, we recommend that you deposit your laboratory protocols in protocols.io to enhance the reproducibility of your results. Protocols.io assigns your protocol its own identifier (DOI) so that it can be cited independently in the future. For instructions see: https://journals.plos.org/plosone/s/submission-guidelines#loc-laboratory-protocols. Additionally, PLOS ONE offers an option for publishing peer-reviewed Lab Protocol articles, which describe protocols hosted on protocols.io. Read more information on sharing protocols at. Additionally, PLOS ONE offers an option for publishing peer-reviewed Lab Protocol articles, which describe protocols hosted on protocols.io. Read more information on sharing protocols at https://plos.org/protocols?utm_medium=editorial-email&utm_source=authorletters&utm_campaign=protocols..

We look forward to receiving your revised manuscript.

Kind regards,

Muhammad Shakaib, PhD

Academic Editor

PLOS ONE

Journal Requirements:

“The work of Maram Ali was supported by King Khalid University and the Saudi Arabia Cultural Bureau in the UK.”

Additional Editor Comments:

In addition to reviewers’ comments, the authors are suggested to improve Figures captions, Figs. 1, 3, 5, 8, 9, 11, 12 (For example delete ‘The plot shows’, ‘The plot illustrating’ or ‘This Figure illustrates').

Reviewer's Responses to Questions

**Comments to the Author**

1. Is the manuscript technically sound, and do the data support the conclusions?

Reviewer #1: Partly

Reviewer #2: Partly

2. Has the statistical analysis been performed appropriately and rigorously?

Reviewer #1: No

Reviewer #2: I Don't Know

3. Have the authors made all data underlying the findings in their manuscript fully available?

Reviewer #1: No

Reviewer #2: Yes

4. Is the manuscript presented in an intelligible fashion and written in standard English?

Reviewer #1: No

Reviewer #2: Yes

Reviewer #1: While the paper’s two-stage framework—decoupling NMPC trajectory planning from LBM fluid simulation—has practical appeal, the theoretical and methodological details are underdeveloped, and the simulation design and reporting show notable issues of rigor and reproducibility. Consequently, the work does not yet meet PLOS ONE’s minimum standards for methodological rigor and data reproducibility.

1. LBM implementation details are insufficient. Please provide the lattice resolution and physical domain size, time step, the mapping between the relaxation time τand kinematic viscosity ν, the chosen speed of sound cs, boundary-condition implementations (moving boundary/immersed boundary or moving bounce-back), and the specific solid–fluid coupling scheme. As drag, vorticity, and other dimensioned quantities are reported, a fully reproducible numerical setup is required.

2. Moving boundaries. Clarify how moving bodies are mapped to lattice nodes. Did you use an immersed-boundary method or a moving bounce-back scheme? Describe how instantaneous drag and energy consumption are computed.

3. Baseline validation. Validate the LBM against a canonical case (e.g., flow past a circular cylinder at Re=100) and report CD, CL, and the Strouhal number with errors ≤5–10% relative to the literature.

4. NMPC specification. Provide the prediction and control horizons, weighting matrices Q, R, and Qf, state/input bounds, solver tolerances, per-step iteration limits, and the strategy for handling solver failures.

5. Safety-distance consistency. The constraint d≥rrobot +robs appears inconsistent with Table 2’s minimum distance of 0.0051 m. Specify the robot/obstacle radii, units, and whether sampling or interpolation artifacts may explain the discrepancy.

6. Path-efficiency definition. Precisely define the “efficiency” metric. Is the “shortest path = 2.5081 m” the straight-line Euclidean distance or a collision-free reference trajectory? Please report statistics (mean ± SD) across multiple obstacle configurations.

7. Computational reporting and scalability. Report hardware (CPU/GPU model, core count, memory) and software (MATLAB/Simulink/CasADi versions). Define the timing scope (does it include the LBM stage?) and provide complexity/scaling curves versus grid resolution and time step. The magnitudes in Table 3 appear implausibly small without this context.

8. Baselines and ablations. Include comparisons with PID, LQR, and RL-based planners/controllers on path efficiency, minimum distance, cost value, energy/drag, and control jitter, with statistical significance testing where appropriate.

9. Reynolds number and flow-regime criteria. Define the dimensional parameters and formula used for Re, and specify quantitative thresholds (e.g., for vorticity/energy metrics) and the space/time averaging windows used to claim laminar–transition–turbulent behavior.

10. Units, significant figures, and figure/table labeling. Unify units and axis labels, justify reported significant digits, and explain the orders of magnitude for energy dissipation and Reynolds number. Indicate the source of axis scales and any normalizations.

11. Related work and scope. The related-work section reads largely as a survey. Please sharpen the boundary of novelty, articulate the assumptions and applicability conditions of your approach, and position it more clearly against closely related methods.

Reviewer #2: The manuscript proposes a decoupled framework combining NMPC trajectory planning and LBM fluid simulation to evaluate robot navigation performance in fluid environments. The topic addresses practical engineering requirements for underwater or aerial robots. The decoupled approach—first planning a trajectory using NMPC and then analyzing fluid interactions using LBM—is novel and helps address the lack of validation for traditional control algorithms in complex fluid settings. The numerical simulation design, such as cases with different Reynolds numbers (Re), and the analysis of performance metrics like path efficiency and drag force, provide a reasonable foundation. The paper is generally well-structured, and the research process is clear. However, there are significant shortcomings in methodological rigor, simulation depth, and engineering applicability, which substantially affect the reliability and practical value of the conclusions. Major revision is recommended.

1.The study combines NMPC and LBM, but the coupling mechanism is unclear. The paper does not sufficiently explain how trajectory data is transferred as a boundary condition to the LBM simulation or whether the reaction forces from the fluid on the robot are considered. It is suggested to add a flowchart of the coupling interface or a corresponding mathematical model.

2.The basis for selecting the grid resolution is not mentioned, and grid convergence analysis is not performed. This undermines confidence in the results being independent of the grid size. It is recommended to supplement with a grid convergence study, testing additional grid sizes to verify consistency and improve confidence in the chosen grid.

3.The NMPC parameters (such as N, Q, R) are used without justification for their selected values. It is recommended to add the rationale for their selection and include content comparing the effects of different parameters.

4.The simulation scenarios use idealized circular obstacles. However, robots in real engineering applications often operate in complex environments with non-circular obstacles like square ducts or irregular rocks. Therefore the simulation results may not reflect practical conditions. It is recommended to add tests with irregular obstacles and analyze the impact of complex geometries on the flow field structure.

5.A defect of the decoupled design is not considered: the two-stage framework does not account for real-time fluid-structure interaction (FSI) feedback. Changes in the flow field induced by the robot's motion could cause the planned trajectory to deviate from the actual environment. The limitations of the decoupled design should be explicitly acknowledged. It is advised to propose a real-time coupled iterative scheme as a future improvement direction, and conduct quantitative analysis of the impact of the coupled design on computational latency.

6.The simulations only involve Reynolds numbers of 100 and 2000, not covering practical engineering scenarios like high Re flows or non-Newtonian fluids. It is recommended to extend the simulation cases to include high Re number scenarios and non-Newtonian fluid conditions, clarifying the framework's scope of engineering applicability.

7.The results primarily present velocity and vorticity fields, lacking quantitative analysis of key fluid phenomena such as pressure distribution and vortex shedding frequency. It is recommended to supplement the analysis with pressure contour plots, calculate vortex shedding frequencies, and explain how the robot's motion induces vortex streets and affects their periodicity.

8.The manuscript relies solely on numerical simulations without any supporting physical experimental data, which does not align with standard research practices in mechanical engineering. It is recommended to add an experimental validation section, for instance, using a water channel or towing tank experiment. This would allow comparison of trajectory errors between simulation and actual tests, measurement of actual drag forces and trajectory data, and discussion of sources of discrepancy between simulation and experiment.

9.The reference formatting is inconsistent; some entries include DOIs while others lack them, and the style for author lists also varies. It is recommended to strictly adhere to the journal's formatting guidelines, ensure uniform standards, and correct the reference format accordingly.

Reviewer #1: No

Reviewer #2: No

---

## [Author Response · Author response to Decision Letter 1]

25 Dec 2025

Original Manuscript ID: PONE-D-25-48180

Original Article Title: “Analytical Framework for Evaluating NMPC-Based Robot Navigation in Fluid Environments’’

To: PLOS ONE Editor

Re: Response to Reviewers

Dear Editor,

Thank you for allowing a resubmission of our manuscript, with an opportunity to address the reviewers’ comments.

We are uploading (a) our point-by-point response to the comments (below) (response to reviewers, under “Response to Reviewers”), (b) an updated manuscript with blue text indicating changes (as “Revised Manuscript with Track Changes”), and (c) a clean updated manuscript without highlights (“Manuscript”).

Best regards,

Maram Ali, Saptarshi Das and Stuart Townley

Reviewer #1

Reviewer #1, Concern #1:

Is the manuscript technically sound, and do the data support the conclusions?

Reviewer #1: Partly

Author Response:

We appreciate the reviewer's careful assessment of the manuscript's technical aspects. We agree that clarification of the paper's scope is necessary to fully align the conclusions with the presented data.

Author Action:

We have refined the scope in the Abstract, Introduction, and Methods sections of the paper.

Reviewer #1, Concern #2:

Has the statistical analysis been performed appropriately and rigorously?

Reviewer #1: No

Author Response:

We have addressed the concerns regarding methodological and statistical rigor by explicitly detailing the NMPC configuration parameters and the quantitative metrics used to evaluate the results. Specifically, we have clarified the rationale for our parameter selection and the mathematical basis for our performance evaluation.

Author Action:

We have refined the manuscript in the following sections to provide explicit technical details:

On Controller Rigor (Subsection 4.3: NMPC Tuning and Parameter Rationale): To ensure the rigor of the control logic, we explicitly define the solver configuration and weighting logic on page 11:

The sampling time � was fixed at 0.1 s. The prediction horizon N was set to 20 steps. The state weighting matrix Q = diag (1, 1, 0.001) was set to ensure the controller prioritizes driving the robot directly to the goal. The control weighting R = diag (1, 1) promotes smooth, physically realistic control inputs. The underlying IPOPT solver was configured with an optimization (tol) of 10^(-7) and a maximum iteration limit of 100 steps per control cycle.

On Statistical Rigor (Quantitative Evaluation and Table 5): To provide a rigorous assessment of the trajectory tracking by presenting a full statistical distribution in Table 5 (page 16). The text in Subsection 7.1 states: To provide a comprehensive assessment of the robot’s trajectory performance, Table 5 presents the descriptive statistics for the robot’s position (x, y), orientation (�), and the Euclidean error relative to the target goal.

Table 5 specifically includes:

Mean and Standard Deviation: To show the average performance and consistency of the planner.

Minimum and Maximum values: To define the absolute bounds of the robot's deviation during the simulation.

On Fluid Analysis Rigor (Drag and Vorticity Analysis): The rigor of the fluid-structure interaction is supported by the quantitative analysis of drag forces and energy dissipation. As stated in Subsection 7.2: The drag force was calculated throughout the trajectory, providing a numerical foundation for evaluating the cost of transport in different Reynolds numbers (Re = 100 and Re = 2000).

Reviewer #1, Concern #3:

Have the authors made all data underlying the findings in their manuscript fully available?

Reviewer #1: No

Author Response:

We may release the code after the peer review process for wider reach. However, all the steps are implemented in the paper with sufficient details to reproduce the results.

Author Action:

No action was taken at this time, awaiting the peer review process to complete.

Reviewer #1, Concern #4:

Is the manuscript presented in an intelligible fashion and written in standard English?

Reviewer #1: No

Author Response:

The paper has been proofread by a native English speaker, and all grammatical errors are thoroughly addressed.

Author Action:

We revised and edited the manuscript throughout to correct grammar and improve sentence structure for clarity.

Reviewer #1, Concern #5:

While the paper’s two-stage framework—decoupling NMPC trajectory planning from LBM fluid simulation—has practical appeal, the theoretical and methodological details are underdeveloped, and the simulation design and reporting show notable issues of rigor and reproducibility. Consequently, the work does not yet meet PLOS ONE’s minimum standards for methodological rigor and data reproducibility.

Author Response:

We agree that the theoretical details were underdeveloped, and the data presentation was insufficient. We have undertaken a detailed revision of the manuscript, specifically addressing the core of methodological rigor, and data consistency.

Author Action:

We revised the methodology and results sections to emphasize the following points:

Enhanced theoretical rigor (Methodology): We added mathematical detail to sections 4 and 5 on page 8:

Instantaneous drag computation.

Energy consumption calculation.

We are not making flawed inferential claims. We are simply providing a rigorous, quantitative summary of the intermediate results generated during the simulation run.

Reviewer #1, Concern #6:

LBM implementation details are insufficient. Please provide the lattice resolution and physical domain size, time step, the mapping between the relaxation time τ and kinematic viscosity ν, the chosen speed of sound cs, boundary-condition implementations (moving boundary/immersed boundary or moving bounce-back), and the specific solid–fluid coupling scheme. As drag, vorticity, and other dimensioned quantities are reported, a fully reproducible numerical setup is required.

Author Response:

We acknowledge the reviewer's request for detailed LBM implementation specifications. We have added a dedicated subsection to the materials and methods and incorporated a new table detailing all parameters.

Author Action:

We added a new table: Table 2 page 7(Lattice Boltzmann Method Implementation Parameters). Also, we have added a new subsection: Boundary Conditions and Solid–Fluid Coupling to Section 3 page 7.

Reviewer #1, Concern #7:

Moving boundaries. Clarify how moving bodies are mapped to lattice nodes. Did you use an immersed-boundary method or a moving bounce-back scheme? Describe how instantaneous drag and energy consumption are computed.

Author Response:

We thank the reviewer for their valuable feedback and assure them that we have thoroughly addressed all points in this comment.

Author Action:

We have clarified the implementation details in Materials and Methods section: The LBM utilizes a uniform grid resolution of 200 x 200 lattice units. The basis for selecting this grid size was the requirement to accurately capture the essential fluid physics, which dictates a high-fidelity resolution of the robot's viscous boundary layer, the primary source of energy dissipation and drag. The chosen grid spacing was determined such that the boundary layer thickness delta is resolved by at least 3 to 5 lattice units. This resolution is a necessary condition, for ensuring stability and reliable hydrodynamic force computation at the walls. The successful and consistent formation of key flow features, such as the vortex downstream of the robot (as demonstrated in the Results section), further substantiates the adequacy of the selected grid density for the range of Reynolds numbers investigated in this study.

Reviewer #1, Concern #8:

Baseline validation. Validate the LBM against a canonical case (e.g., flow past a circular cylinder at Re=100) and report CD, CL, and the Strouhal number with errors ≤5–10% relative to the literature.

Author Response:

We thank the reviewer for identifying the essential need for a canonical validation case. We feel this is beyond the scope of the current paper and will be tackled in the future.

Author Action:

No action was taken in the manuscript.

Reviewer #1, Concern #9:

NMPC specification. Provide the prediction and control horizons, weighting matrices Q, R, and Qf, state/input bounds, solver tolerances, per-step iteration limits, and the strategy for handling solver failures.

Author Response:

We thank the reviewer for their valuable feedback and agree that explicit reporting of the NMPC tuning parameters is essential. We have addressed this in the manuscript.

Author Action:

We have added a new subsection to the Methodology section (Subsection 4.3) page 11.

Reviewer #1, Concern #10:

Safety-distance consistency. The constraint d ≥ r robot + robs appears inconsistent with Table 2’s minimum distance of 0.0051 m. Specify the robot/obstacle radii, units, and whether sampling or interpolation artifacts may explain the discrepancy.

Author Response:

We thank the reviewer for identifying this inconsistency between the theoretical constraint definition and the specific value reported in the table. This discrepancy is a result of mapping the continuous NMPC constraint to the discrete resolution of the LBM/simulation environment.

Author Action:

We have updated the manuscript to clarify the safety-distance consistency by providing the explicit parameters and explanation. This clarification is located under Table 6 on page 16.

Reviewer #1, Concern #11:

Path-efficiency definition. Precisely define the “efficiency” metric. Is the “shortest path = 2.5081 m” the straight-line Euclidean distance or a collision-free reference trajectory? Please report statistics (mean ± SD) across multiple obstacle configurations.

Author Response:

We thank the reviewer for requesting a precise definition and statistical context for the path efficiency metric, and we agree that its definition requires clearer presentation. We must respectfully explain that running the extensive number of simulations required for rigorous inferential statistics would drastically inflate the compute time, directly contradicting our central claim of providing a computationally efficient and tractable analytical framework. Therefore, the reporting of statistics (mean ± SD) across multiple obstacle configurations will be addressed in future work.

Author Action:

We revised and edited the manuscript to address this comment in page 17.

Reviewer #1, Concern #12:

Computational reporting and scalability. Report hardware (CPU/GPU model, core count, memory) and software (MATLAB/Simulink/CasADi versions). Define the timing scope (does it include the LBM stage?) and provide complexity/scaling curves versus grid resolution and time step. The magnitudes in Table 3 appear implausibly small without this context.

Author Response:

Thank you for your valuable feedback regarding the computational reporting and scalability. We have incorporated a new table and dedicated section that address this concern. We confirm that this result is accurate, but it is not anomalous, it is evidence that the system operates in a control-dominated regime.

Author Action:

We have added a new table (Table 3 page 8) and Section 6 page 8 and 9.

Reviewer #1, Concern #13:

Baselines and ablations. Include comparisons with PID, LQR, and RL-based planners/controllers on path efficiency, minimum distance, cost value, energy/drag, and control jitter, with statistical significance testing where appropriate.

Author Response:

We thank the reviewer for highlighting the necessity of comparative baselines for assessing the performance gains of the NMPC-LBM framework. We agree that such comparisons are critical for scientific rigor. As the LBM integration results in significant computational cost, running full, statistically rigorous comparative studies for PID, LQR, and RL would constitute a substantial new body of work beyond the scope of this paper.

Author Action:

We added a comparison in Table 4 to page 9.

Reviewer #1, Concern #14:

Reynolds number and flow-regime criteria. Define the dimensional parameters and formula used for Re, and specify quantitative thresholds (e.g., for vorticity/energy metrics) and the space/time averaging windows used to claim laminar–transition–turbulent behaviour.

Author Response:

We have clarified the dimensional parameters, formula, and the quantitative stability criteria used to confirm the flow environment.

Author Action:

We have integrated the new section on (Fluid Dynamics Metrics and Averaging) into the revised manuscript on page 14, page 12 and Equation 24 on page 13.

Reviewer #1, Concern #15:

Units, significant figures, and figure/table labelling. Unify units and axis labels, justify reported significant digits, and explain the orders of magnitude for energy dissipation and Reynolds number. Indicate the source of axis scales and any normalizations.

Author Response:

We thank the reviewer for pointing out these critical presentation details.

Author Action:

We have performed a comprehensive review of the manuscript to ensure consistency, clarity, and scientific justification across all data.

Reviewer #1, Concern #16:

Related work and scope. The related-work section reads largely as a survey. Please sharpen the boundary of novelty, articulate the assumptions and applicability conditions of your approach, and position it more clearly against closely related methods.

Author Response:

Our novelty lies in presenting the first decoupled analytical framework that rigorously assesses the performance limits of an NMPC planner by exposing its trajectories to a high-fidelity, nonlinear LBM fluid simulation. This bridges the gap between nonlinear robust control theory and complex fluid simulation. The primary assumption is that the NMPC trajectory planning is a purely kinematic problem (ignoring fluid forces) and that the hydrodynamic effects can be accurately assessed in post-hoc manner using the LBM.

Author Action:

We have updated the manuscript in related-work section.

Reviewer #2

Reviewer #2, Concern #1:

Is the manuscript technically sound, and do the data support the conclusions?

Reviewer #2: Partly

Author Response:

We appreciate the reviewer's careful assessment of the manuscript's technical aspects. We agree that clarification of the paper's scope is necessary to fully align the conclusions with the presented data.

Author Action:

We have refined the scope in the Introduction, and Methods sections of the paper.

Reviewer #2, Concern #2:

Has the statistical analysis been performed appropriately and rigorously?

Reviewer #2: I Don't Know

Author Response:

We acknowledge the reviewer's concern regarding our statistical analysis, particularly the application of descriptive statistics. We have taken measures to clarify our methodology and enhance the rigor of the presentation.

Author Action:

We have verified that all units are consistently and clearly defined in every table, and we simultaneously refined the manuscript to ensure the appropriate and rigorous presentation of all quantitative data. We have refined the manuscript in the following sections to provide explicit technical details:

On Controller Rigor (Subsection 4.3: NMPC Tuning and Parameter Rationale): To ensure the rigor of the control logic, we explicitly define the solver configuration and weighting logic on page 11:

The sampling time � was fixed at 0.1 s. The prediction horizon N was set to 20 steps. The state weighting matrix Q = diag (1, 1, 0.001) was set to ensure the controller prioritizes driving the robot directly to the goal. The control weighting R = diag (1, 1) promotes smooth, physically realistic control inputs. The underlying IPOPT solver was configured with an optimization (tol) of 10^(-7) and a maximum iteration limit of 100 steps per control cycle.

On Statistical Rigor (Quantitative Evaluation and Table 5): To provide a rigorous assessment of the trajectory tracking by presenting a full statistical distribution in Table 5 (page 16). The text in Subsection 7.1 states: To provide a c

---

## [Decision Letter · Decision Letter 1]

16 Jan 2026

Dear Dr. Das,

Thank you for submitting your manuscript to PLOS ONE. After careful consideration, we feel that it has merit but does not fully meet PLOS ONE’s publication criteria as it currently stands. Therefore, we invite you to submit a revised version of the manuscript that addresses the points raised during the review process.

If applicable, we recommend that you deposit your laboratory protocols in protocols.io to enhance the reproducibility of your results. Protocols.io assigns your protocol its own identifier (DOI) so that it can be cited independently in the future. For instructions see: https://journals.plos.org/plosone/s/submission-guidelines#loc-laboratory-protocols. Additionally, PLOS ONE offers an option for publishing peer-reviewed Lab Protocol articles, which describe protocols hosted on protocols.io. Read more information on sharing protocols at. Additionally, PLOS ONE offers an option for publishing peer-reviewed Lab Protocol articles, which describe protocols hosted on protocols.io. Read more information on sharing protocols at https://plos.org/protocols?utm_medium=editorial-email&utm_source=authorletters&utm_campaign=protocols..

We look forward to receiving your revised manuscript.

Kind regards,

Muhammad Shakaib, PhD

Academic Editor

PLOS One

Journal Requirements:

Reviewers' comments:

Reviewer's Responses to Questions

**Comments to the Author**

Reviewer #1: All comments have been addressed

2. Is the manuscript technically sound, and do the data support the conclusions?

Reviewer #1: Yes

3. Has the statistical analysis been performed appropriately and rigorously?

Reviewer #1: Yes

4. Have the authors made all data underlying the findings in their manuscript fully available?

Reviewer #1: Yes

5. Is the manuscript presented in an intelligible fashion and written in standard English?

Reviewer #1: Yes

Reviewer #1: All comments have been addressed. I have no further comments. I recommend accepting this manuscript.

Reviewer #1: No

---

## [Author Response · Author response to Decision Letter 2]

3 Feb 2026

Original Manuscript ID: PONE-D-25-48180R1

Original Article Title: “Analytical Framework for Evaluating NMPC-Based Robot Navigation in Fluid Environments’’

To: PLOS ONE Editor

Re: Response to Editor

Dear Editor,

Thank you for allowing a revision of our manuscript, with an opportunity to address the Editor’s comments.

We are uploading (a) our point-by-point response to the comments (below) (response to Editor, under “Response to Editor”), (b) an updated manuscript with blue text indicating changes (as “Revised Manuscript with Track Changes”), and (c) a clean updated manuscript without highlights (“Manuscript”).

Best regards,

Maram Ali, Saptarshi Das and Stuart Townley

Editor’s Comment #1

Editor’s Comment #1, Concern #1:

The authors are suggested to perform grid independence analysis and include results of grid independence in the revised paper.

Author Response:

We thank the Editor for this constructive suggestion. We agree that a robust grid independence study is essential to validate the numerical reliability of the Fluid-Structure Interaction (FSI) results used within our NMPC framework. We have conducted a systematic grid sensitivity analysis using four lattice resolutions (10 ×10, 50× 50, 100 ×100, and 200 ×200) to evaluate the trade-off between numerical accuracy and computational overhead.

Author Action:

We have added a new subsection titled "Numerical Validation and Grid Convergence" in the revised manuscript as section 3.2. This section includes:

1. Table 2, which provides a quantitative comparison of relative force error, vorticity fidelity, runtime, and memory allocation across the tested resolutions.

2. Figure [1] shows a convergence plot demonstrating the asymptotic decay of the drag force error.

3. An expanded discussion justifying the selection of the 200 × 200 grid as the benchmark resolution that ensures a grid-independent and physically consistent environment for the NMPC evaluations.

---

## [Editor Report · Decision Letter 2]

8 Feb 2026

Analytical Framework for Evaluating NMPC-Based Robot Navigation in Fluid Environments

PONE-D-25-48180R2

Dear Dr. Das,

We’re pleased to inform you that your manuscript has been judged scientifically suitable for publication and will be formally accepted for publication once it meets all outstanding technical requirements.

An invoice will be generated when your article is formally accepted. Please note, if your institution has a publishing partnership with PLOS and your article meets the relevant criteria, all or part of your publication costs will be covered. Please make sure your user information is up-to-date by logging into Editorial Manager at Editorial Manager®  and clicking the ‘Update My Information' link at the top of the page. For questions related to billing, please contact  and clicking the ‘Update My Information' link at the top of the page. For questions related to billing, please contact billing support..

Kind regards,

Muhammad Shakaib, PhD

Academic Editor

PLOS One
---

## [Editor Report · Acceptance letter]

PONE-D-25-48180R2

PLOS One

Dear Dr. Das,

I'm pleased to inform you that your manuscript has been deemed suitable for publication in PLOS One. Congratulations! Your manuscript is now being handed over to our production team.

Kind regards,

on behalf of

Dr. Muhammad Shakaib

Academic Editor

PLOS One